# Genomic landscape of adult testicular germ cell tumours in the 100,000 Genomes Project

Máire Ní Leathlobhair [1,2,3] ✉, Anna Frangou [4,23], Ben Kinnersley [5,6,23], Alex J. Cornish [5], Daniel Chubb[5], Eszter Lakatos [7], Prabhu Arumugam[8], Andreas J. Gruber [9], Philip Law [5], Avraam Tapinos[10], G. Maria Jakobsdottir [11,12], Iliana Peneva[1], Atef Sahli[1,11], Evie M. Smyth [3], Richard Y. Ball [13], Rushan Sylva[14], Ksenija Benes[15], Dan Stark[16], Robin J. Young[17], Alexander T. J. Lee[12], Vincent Wolverson[8], Richard S. Houlston [5], Alona Sosinsky [8], Andrew Protheroe [18,24], Matthew J. Murray [19,20,24] ✉, David C. Wedge [1,11,12,24] ✉, Clare Verrill [21,22,24] ✉, Testicular Cancer Genomics England Clinical Interpretation Partnership Consortium* & Genomics England Research Consortium*

Testicular germ cell tumours (TGCT), which comprise seminoma and non-seminoma subtypes, are the most common cancers in young men. In this study, we present a comprehensive whole genome sequencing analysis of adult TGCTs. Leveraging samples from participants recruited via the UK National Health Service and data from the Genomics England 100,000 Genomes Project, our results provide an extended description of genomic elements underlying TGCT pathogenesis. This catalogue offers a comprehensive, high-resolution map of copy number alterations, structural variation, and key global genome features, including mutational signatures and analysis of extrachromosomal DNA amplification. This study establishes correlations between genomic alterations and histological diversification, revealing divergent evolutionary trajectories among TGCT subtypes. By reconstructing the chronological order of driver events, we identify a subgroup of adult TGCTs undergoing relatively late whole genome duplication. Additionally, we present evidence that human leukocyte antigen loss is a more prevalent mechanism of immune disruption in seminomas. Collectively, our findings provide valuable insights into the developmental and immune modulatory processes implicated in TGCT pathogenesis and progression.

Although testicular germ cell tumours (testicular GCT; TGCT) are rare, they are the most common malignancy in young men, with the highest incidence occurring in those aged 30–34 years[1]. The two primary histological types of TGCT are seminoma and non-seminomatous germ cell tumours (NSGCT). NSGCT are typically more aggressive than seminomas, and comprise the histological subtypes embryonal carcinoma (EC), yolk sac tumour (YST), and choriocarcinoma, as well as teratoma[2]; often multiple histologies can co-exist within a single lesion.

A full list of affiliations appears at the end of the paper.  *Lists of authors and their affiliations appear at the end of the paper.

✉e-mail: mnileathlobhair@gmail.com; mjm16@cam.ac.uk; david.wedge@manchester.ac.uk; clare.verrill@ouh.nhs.uk

Unlike most other cancers, TGCTs are rarely caused solely by somatic driver mutations, but arise from failure to control the latent developmental potential of their cell-of-origin, a foetal germ cell, resulting in its reprogramming[3].

There is increasing evidence that the clinical behaviour of cancer and therapeutic response reflects underlying tumour genomics. Thus far, genome sequencing studies of TGCT[4–7] have mostly been confined to examining the protein-coding (exome) sequence[4,5,8,9] or small whole genome sequenced cohorts[10], and a comprehensive description of the whole genomic landscape of TGCT is largely missing from current literature. Notably, the largest whole genome sequencing (WGS) study of TGCT reported to date was based on the sequencing of only nine postpubertal (age > 12 years) patients[11]. There has been limited exploration of key mutational processes, such as structural variation, and their signatures in TGCT, and much remains to be explored regarding germ cell tumour evolution. To advance our understanding of TGCT, we examined 60 whole genome sequenced cancers from 57 TGCT patients, recruited from seven National Health Service (NHS) Genomic Medicine Centres across England, sampled as part of the Genomics England (GEL) 100,000 Genomes Project (100kGP) [GEL v12 data release][12,13]. In this work, we provide an extensive analysis of the genomic landscape, mutational processes, and clonal architecture underlying the development of adult TGCT.

## Results
### Overview of the Genomics England TGCT cohort
All TGCT tumour-normal sample pairs were processed through 100kGP bioinformatic somatic-variant analysis pipelines (tumour coverage: 95–122.7×, mean: 108.7×; normal: 31.3–62.6×, mean: 39.3×). We restricted our analysis to high-quality data derived from fresh frozen material, involving 60 tumour samples from 57 individuals [age 17–77 years (y); median 35 y]; (55 untreated primary and five late-stage treated metastatic TGCT), including four primary tumour regions sampled from a single participant (Fig. 1a, Supplementary Data 1). The primary tumours comprised 39 pure seminomas and 16 NSGCT, including three EC cases and one undifferentiated teratoma. A bimodal age distribution was observed at diagnosis in participants, as expected, with most seminomas being diagnosed between age 20 y and 40 y (Supplementary Fig. 1).

Across the GEL cohort, we identified 80,760 individual single nucleotide variants (SNVs), 7412 small insertions and deletions (indels), and 1865 chromosomal rearrangements (Fig. 1b, Methods). As per previous reports, tumours were typified by a uniformly low rate of single nucleotide variants (SNVs; mean genome-wide substitution rate of 0.475/Mb; range 0.095–1.62), likely reflecting the embryological origins of TGCT[4,5]. No tumour displayed a hypermutated phenotype, i.e., excessively high SNV/indel mutation burden (maximum SNV/Mb = 1.62; maximum indel/Mb = 0.28).

### Identifying subtypespecific driver mutations
Using the IntOGen pipeline, which aggregates seven complementary driver discovery algorithms, we searched for driver genes across the GEL TGCT cohort (Methods, Supplementary Data 2). Eight genes were significantly somatically mutated (*KIT*, *KRAS*, *NRAS*, *RAC1*, *SPEN*, *EP300*, *KLF4*, *KMT2C*). Consistent with The Cancer Genome Atlas (TCGA) study of TGCT[5], *KIT* driver mutations defined a subset of seminomas (Fig. 2). Mutations in *KIT* were clustered primarily in exon 17, in a pattern similar to that previously reported in testicular seminomas and intracranial GCTs (Supplementary Fig. 2, Supplementary Data 2)[14,15]. Multiple mutations affecting the same oncogene (*KIT*) were observed in only one participant with a clinical stage II seminoma. Further analysis identified additional drivers defining distinct subgroups within the seminoma subtype. These encompassed gain-of-function mutations in the transcription factor *KLF4* and the GTPase *RAC1*, as well as loss-of-function

mutations in the histone acetyltransferase *EP300* (Fig. 2). Additionally, we searched for non-coding drivers using three complementary algorithms, namely OncodriveFML[16], OncodriveCLUSTL[17], and ActiveDriverWGS[18]. However, we did not identify any significant non-coding elements under positive selection (Supplementary Fig. 3).

To supplement our analysis of GEL tumours, we reanalysed data from TGCT cohorts within the TCGA[5] (128 samples) and Memorial Sloan Kettering - Metastatic Events and Tropisms (MSK-MET)[19] (128 samples) studies, allowing us to identify further subtype-specific coding drivers. Within the TCGA dataset, somatic mutations in 10 genes reached significance, including *NOTCH1*, *PIK3CA*, *BIRC6*, *ARID1B*, and *LRP1B*. Two putative driver genes, *PTMA* and *FAT4*, were identified in NSGCT subtypes and primarily subject to loss-of-function mutations (Supplementary Data 2). GEL cohort data also provided support for *PTMA* (*prothymosin alpha*) as a putative driver gene, previously implicated in TGCT, though not currently included in the COSMIC Cancer Gene Census[20,21].

Finally, we assessed the clinical actionability of identified driver gene mutations by referencing the OncoKB Knowledge Base (http://oncokb.org/)[22], and found that 17% (19/110) of alterations annotated by OncoKB were targetable (OncoKB Level 1-4). Most targetable mutations (18/19) were Level 3B, indicating predictive biomarkers that are considered standard-of-care for a different tumour type.

### Cancer driver genes in focal genomic alterations
The Battenberg algorithm was used to estimate clonal and subclonal copy number variation across the cohort[23]. Applying GISTIC2[24] to these profiles, we identified 29 genomic regions recurrently affected by focal amplifications and deletions (Methods, Supplementary Fig. 4, Supplementary Data 3). In addition to established recurrent copy number alterations (CNAs), including chromosome arm-level gains spanning *KRAS* (12p), amplifications involving *KIT* (4q12; 19% cases) and *MDS2* (1p36.32; 17% cases), and deletions spanning *DMRT1* (9p24.3; 37% cases), which is associated with testicular germ cell tumour susceptibility[25], we identified 26 additional novel events. Although *KIT* mutations appeared to be restricted to a subset of seminomas, amplifications spanning *KIT* were also observed in NSGCT (Fig. 2). Segments 1q21.3 (14% cases), 7q11.23 (46% cases), and 22q11.1 (25% cases) spanning oncogenes *SETDB1*, *CDK6*, and *DGCR8* respectively, were found to be recurrently amplified. Focal deletions spanning cyclin A1 (*CCNA1*) and the transcription factor *FOXO1* (13q13.3), critical for successful spermatogenesis, were found to occur exclusively in seminomas. Notably, focal gains spanning AFP (4q13.2) were also restricted to a subset of seminomas (8/57). Although alpha-fetoprotein is a serum tumour marker typically associated with non-seminomatous germ cell tumours, previous reports have noted elevated serum AFP levels in some cases of histologically pure seminomas[26,27]. Several recurrent deletions spanned WNT signalling-related genes including the cadherins *CDH*1 and *CDH11*, *CREBBP* (16q24.2) and *SMAD4* (18q22.2). Mutual exclusivity analysis revealed that the most prominent driver events were largely not co-occurring, although the most significant driver interactions identified were cooperating events including *PIK3CA-MCL1* amplifications, and *RB1-FLI1*, *RB1-MEN1*, and *MAF-SMAD4* deletions (Supplementary Fig. 5). Mutually exclusive events were identified involving *KIT-MAF* and *PTMA-MCL1*. The sole intra-chromosomal pair identified consisted of co-occurring *MEN1-FLI1* deletions.

A primary somatic feature in TGCT development is copy number gain of chromosome 12p, typically structured as an isochromosome (i12p)[7,28]. We observed allelic copy number profiles consistent with the presence of at least one i12p in 75% (43/57) of tumours. A subset of these (5/43; 12%) were categorised as canonical chromosomes (Supplementary Methods) but characterised by complex rearrangements of the 12p arm. Complex i12p cases were all seminomas with

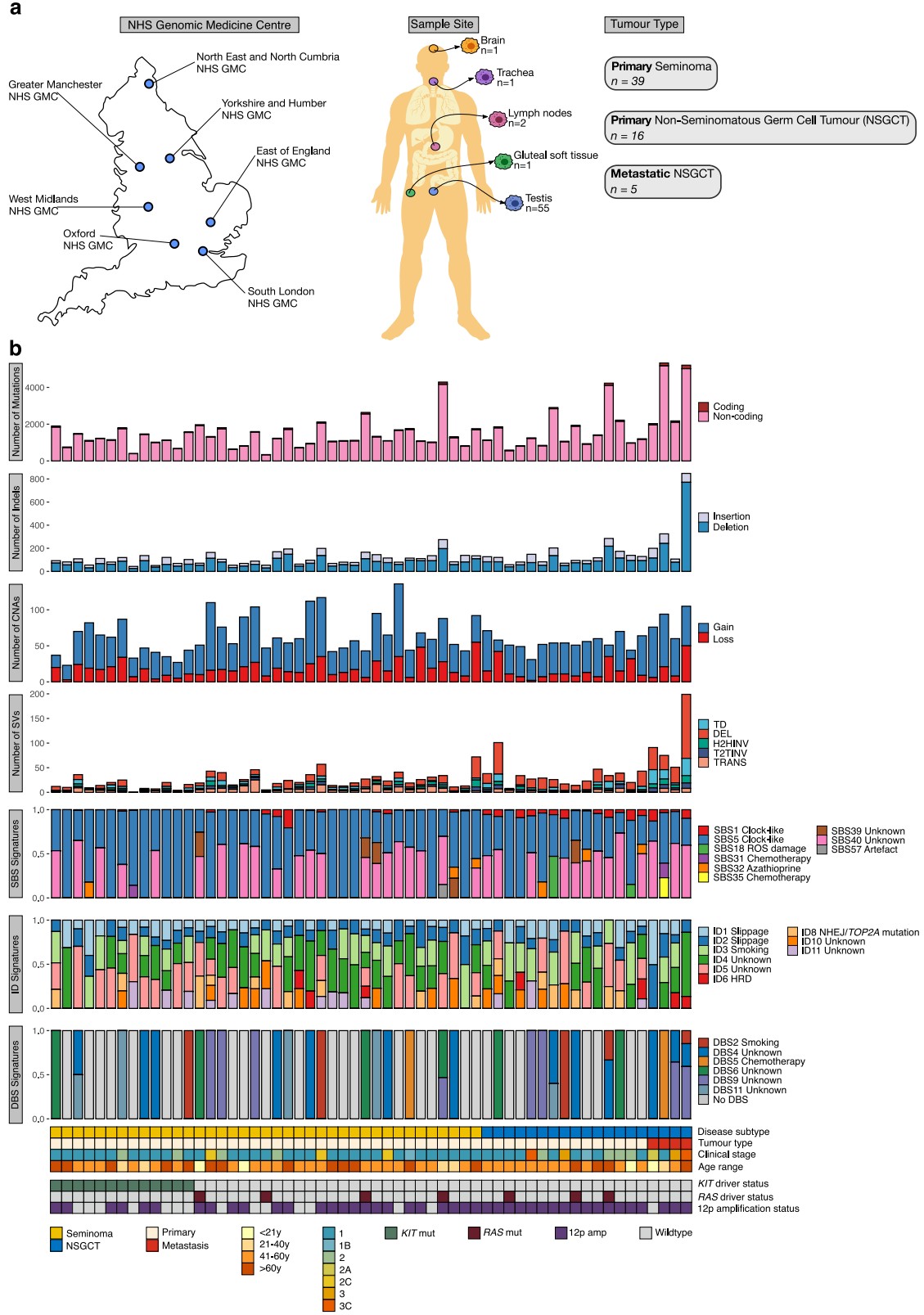

recurrent focal loss at 11q24.3 encompassing the ETS transcription factor *FLI1* (Supplementary Data 3). Most tumours lacking the i12 p event were seminomas (13/14; 93%) and instead had at least four copies of 12p. Only two samples exhibited 12q loss of heterozygosity (LOH), suggesting that most tumours had undergone duplication of chromosome 12 or a second WGD before i12p formation, as previously described[29].

## Hotspots of structural variation in TGCT

Using methods described by Glodzik et al.[30], we identified a single structural variant hotspot involving large (>100 kilobases, kb) tandem duplications (TD) and eight deletion hotspots (Supplementary Data 4). We observed one TD hotspot in the region of chr19:55–58 Mb spanning the histone methyltransferase, GLP. Interestingly, a gain-of-function mutation in the *Caenorhabditis elegans* Notch receptor glp-1 has been

**Fig. 1 | Mutational landscape of adult testicular germ cell tumours (TGCT).**
**a** Genomic profiling of primary and metastatic TGCT samples with matched germline DNA from peripheral blood. Samples were collected from 60 participants recruited from seven NHS Genomic Medicine Centres (GMCs) across England as indicated on the map. The human silhouette drawing was modified from a template from V<underline>ecteezy.com</underline> (https://www.vecteezy.com/vector-art/299365-medical-infographic-of-human-body). **b** From top to bottom: number of coding mutations identified in each sample; number of insertions and deletions (indels) in each sample; total number of structural variants in each sample, separated into tandem duplications (TD), deletions (DEL), head-to-head (H2HINV) and tail-to-tail (T2TINV) inversions, transversions (TRANS); proportion of mutations

assigned to single base substitution (SBS), insertion/deletion (ID), and doublet base substitution (DBS) mutational signatures; TGCT subtype; tumour type (primary or metastasis); clinical stage; age-group of participant; mutation status of *KIT* driver gene; mutation status of *RAS* (*KRAS* or *NRAS*) driver genes; presence or absence of 12p amplification according to GISTIC2. Exposures or processes linked with mutational signatures are listed. Two samples that were not sequenced via a PCR-free workflow are excluded from this figure. HRD homologous recombination deficiency, amp amplification, mut driver mutation, NHEJ non-homologous end joining, NSGCT non-seminomatous germ cell tumours, ROS reactive oxygen species, y years of age.

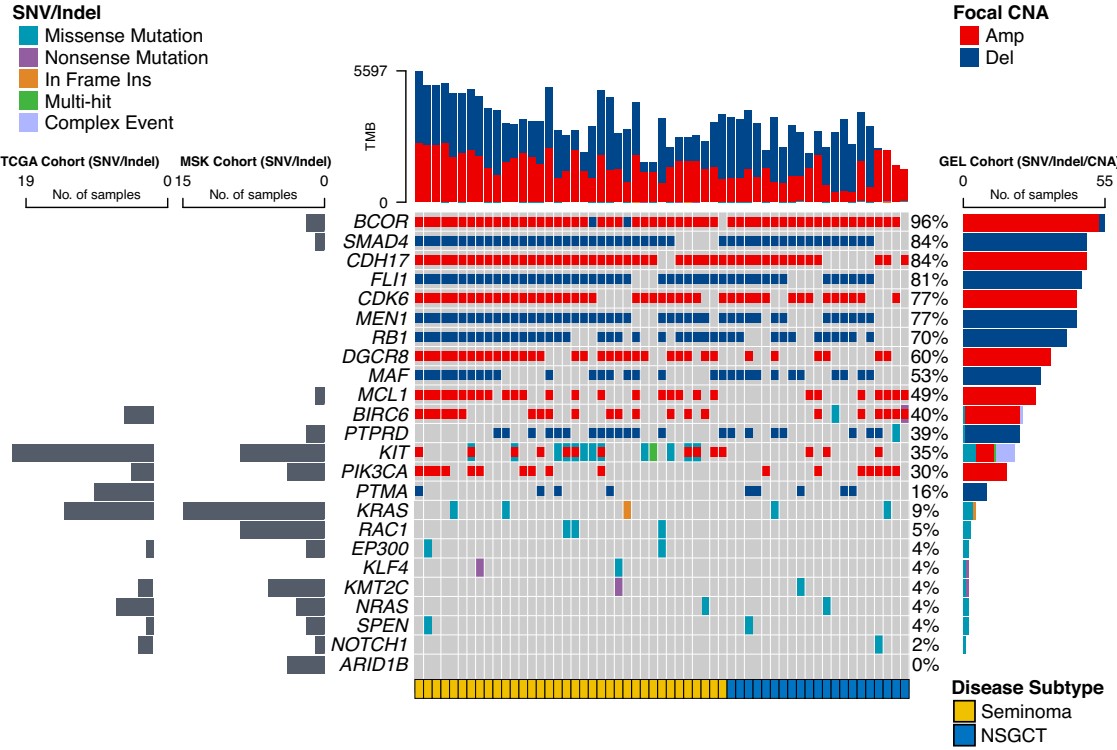

**Fig. 2 | Heatmap of molecular mutations in testicular germ cell tumours (TGCT).** In total, 57 individual adult participant samples were analysed. Point mutation and indel drivers independently identified in The Cancer Genome Atlas (TCGA) and Genomics England (GEL) TGCT cohorts are shown alongside annotated GISTIC2 focal segments. Driver presence/absence in the Memorial Sloan Kettering −Metastatic Events and Tropisms (MSK-MET) TGCT cohort is also shown. Seventeen recurrently mutated genes were found in the cohort, respectively, with *KIT*

being the most frequently altered gene (Supplementary Data 2). The colour code indicates mutation type(s) or TGCT subtype (see legends). Samples with more than one type of mutation (missense, nonsense or in-frame insertion) in the same gene correspond to 'Multi Hit' events. Samples with driver mutations and amplifications/deletions in the same gene are indicated with an overlay of two colours. Amp amplification, Del deletion, CNA copy number alteration, NSGCT non-seminomatous germ cell tumours.

described, leading to germline tumour formation[31]. However, this hotspot did not overlap with any GISTIC-defined focal amplifications. Deletion hotspots associated with copy number loss were centred on chr3:60 Mb spanning the fragile histidine triad (*FHIT*) gene, chr9:7-12 Mb covering the tyrosine phosphatase *PTPRD*, and chr16: 78–84 Mb targeting cadherin 13 (*CDH13*). We observed chromothripsis in one tumour (GEL-TGCT-0056), a rare case of metastatic teratoma with somatic-type malignancy, in which a cluster of 23 structural variants arose in a single catastrophic event affecting chromosomes 7 and 17, including amplification of *PPM1D* (Supplementary Data 4). Canonical translocations and fusions associated with Ewing's sarcoma and related primitive neuroectodermal tumours were not detected in this participant.

## KRAS amplification on extrachromosomal DNA

We next leveraged the GEL dataset to explore the landscape of extrachromosomal DNA (ecDNA) formation in testicular cancer. EcDNA is

often associated with oncogene amplification and poor clinical outcomes in many cancers[32]. Amplicon structures were detected and classified in TGCT using the Amplicon Architect tool[33]. Amplicons were identified in 85% (46/54) of the TGCT samples (Methods, Supplementary Data 5). The size of single-interval amplicons detected ranged from 116 kb to 76 Mb (median 4 Mb), and over 85% (113/130) were >1 Mb. Complex rearrangements identified in at least two samples spanned the established TGCT oncogenes *KRAS*, *MYC*, *EGFR*, and members of signalling pathways commonly dysregulated in cancers including WNT (*SOX2*), RTK (*PDGFRA*), and the p53 pathway inhibitors, *CDK4/6* and *MDM2* (Supplementary Data 5). The only oncogene identified within cyclic amplicon structures, including ecDNA in one instance, was *KRAS*, and only in seminomas. Amplicons showing a signature[34] of having been created by a breakage-fusion-bridge mechanism were also exclusively identified in seminomas. Seminomas also carry a significantly higher number of amplicon structures relative to NSGCT (*p* = 0.018; Supplementary Fig. 6).

## Complete repertoire of mutational signatures

To gain insight into the aetiological basis of mutation, we extracted mutational signatures (Supplementary Figs. 7–8, Supplementary Data 6). In most tumours, the majority of single base substitutions (SBS) could be assigned to signatures SBS5/SBS40 and SBS1 (using nomenclature established in ref.[35]), thought to result from endogenous clock-like mutagenic processes (Supplementary Fig. 8); however only SBS5 and the number of C > T mutations at NpCpG trinucleotides correlated with age ($p = 4.3 \times 10^{-8}$ and $p = 0.02$, respectively; Supplementary Fig. 9). Seminomas with mutant *KIT* had significantly lower SBS1 than either wild-type seminomas ($p = 0.0028$) or NSGCT ($p = 5.5 \times 10^{-6}$).

Some TGCT subtypes exhibited distinct SBS patterns. SBS18, a signature linked with damage by reactive oxygen species (ROS), was detected in two tumours, both NSGCT with minor YST components. Notably, in GEL-TGCT-0038, the majority of variants were attributable to SBS18. This signature has previously been described in multiple paediatric cancers, placental tissue, and most recently in patients with pre- and peripubertal YSTs[11,36]. A signature attributable to platinum chemotherapy exposure, SBS35, was detected in two post-chemotherapy metastases, as expected. SBS31, another signature related to platinum drug treatment, was also found in a clinical stage I primary seminoma treated with radical orchiectomy and carboplatin after sampling. SBS32, a signature not reported in prior TGCT studies, and associated with azathioprine treatment[37], was detected in 11% (6/57) of participants, despite no documented medical history indicating that any of these participants had received such treatment. Of note, a similar finding was recently reported in acute myeloid leukaemia patients, implying mutational mechanisms other than exposure to azathioprine may contribute to SBS32[38]. Changes in mutational signature activity between clonal and subclonal mutations were observed (Supplementary Fig. 10), with a general trend towards a lower proportion of subclonal mutations attributed to SBS5 ($p = 2.2 \times 10^{-16}$; test for trend in proportions). Analysis of the indel (ID) mutational spectra revealed a predominance of ID1 and ID2, both due to slippage during DNA replication[35]. Deletion patterns characterised by ID6 and ID8 and arising from distinct mechanisms of DNA double-strand break repair[35] were mutually exclusive (Supplementary Data 6). The majority of doublet base signatures (DBS) identified in TGCT were of unknown aetiology, except for those associated with tobacco smoking (DBS2) and platinum chemotherapy (DBS5).

We next examined mutational processes generating genomic rearrangements in TGCT. To detect these, we first applied a recently developed framework[39] for classifying chromosomal instability in cancer from 21 pan-cancer copy number signatures (CN1-CN21) (Supplementary Figs. 11–12). The tetraploidy-associated signature CN2 was found in most samples, across both seminomas and NSGCT. We also identified an attribution of both CN1 and CN2 signatures together across a number of tumours, indicating a hyperdiploid or sub-tetraploid profile[39]. We identified contributions from CN13-CN15, a family of signatures characteristic of specific numerical chromosomal instability, encompassing whole-arm or whole-chromosome-scale loss of heterozygosity events. CN13, which is dominated by LOH segments of total copy number 1, was restricted to NSGCT. Co-occurrence of signatures CN1, CN13, and CN15 was observed in a small number of participants (3/57; 5%) with copy number profiles showing significant amounts of copy-neutral LOH and only in metastatic samples or, notably, primary cases that reported subsequent metastases, suggesting potential clinical relevance for this signature in TGCT.

Next, we classified structural rearrangements in subclasses considering their type and size (Methods), applying the same statistical framework used for other classes of mutational signatures[40]. This approach revealed two structural variant signatures (S1, S2) (Supplementary Figs. 13–14), present in both seminomas and NSGCT. Signatures S1 and S2 were similar to recently described rearrangement reference signatures characterised by unclustered translocations (RefSig R2) and unclustered deletions up to 100 kb (RefSig R5), respectively (Supplementary Fig. 15)[40]. Previously described associations include RefSig R5 with *BRCA2* mutations and RefSig R2 with driver mutations in *TP53*[40]. Although *BRCA2* mutations were not detected in the GEL cohort, tumours exhibiting recurrent deletions spanning *BRCA2* displayed a significantly higher prevalence of signature S2 rearrangements ($p = 0.001528$, Wilcoxon rank sum test). Other signatures associated with inefficient homologous recombination repair are either not detected in the GEL TGCT cohort (SBS3) or are present in a small number of cases (ID6/ID8). Thus, it is not clear that loss of *BRCA2* contributes to the overall signature repertoire.

## Prevalence of whole genome duplication

Whole genome duplications (WGD) are near universal in TGCT, with recent work showing these events occur early in embryogenesis[5,11]. In all but one case (56/57), tumours from the GEL cohort were shown to have undergone WGD (Supplementary Fig. 16). Using MutationTimeR[41], we timed somatic mutations relative to copy number gains and calculated the relative timing of these gains. We then timed the occurrence of WGD, using the ratio of clock-like mutations occurring before and after WGD (Methods). We observed a median of ~9 substitutions (range 0–375) occurring prior to WGD, and in seven cases we did not observe any pre-WGD substitutions, supporting early occurrence of genome duplication, likely *in utero* (Fig. 3a). This observation is in stark contrast to most solid cancers, where WGD events are broadly distributed throughout clonal evolution and likely stochastic (Fig. 3b). However, in three cases, genome doubling events were estimated to occur much later relative to the rest of the cohort. One of these samples, an extensively metastatic GCT with a predominant EC component, carried an estimated 375 pre-duplication substitutions. A further two cases, both clinical stage I seminomas, also exhibited relatively late WGD. Both were metachronous bilateral testicular tumours; one participant had their first TGCT diagnosis almost 30 years before 100kGP sample collection, and the other was diagnosed for a second time five years after sampling. In the metastatic case, there was a past history of a bilateral retractile testis but with no previous report of bilateral TGCT. Recent single cell analyses suggest that neonates possess a small pool of gonadal cells with characteristics of primordial germ cells (PGCs) in their testes[42]. It is therefore conceivable that PGC-like cells lingering into infancy could undergo the same WGD process.

We then estimated the time point during PGC development that WGD occurred by dividing pre-duplication substitution burden estimates by the reported mutation rate per cell division within PGCs[43], as described in Oliver et al.[11] (Supplementary Methods). Excluding the late WGD cases, median WGD was estimated to occur at ~11 cell divisions in TGCT (range 0–71.5, lower and upper bounds of post-PGC cell divisions), setting the genetic hallmark of TGCT initiation in the developmental period. Most tumours with WGD (42/56; 75%) had synchronous chromosomal gains (Supplementary Methods, Fig. 3c), broadly in line with the distribution of gain patterns reported by the Pan-Cancer Analysis of Whole Genomes (PCAWG) in tumours with WGD[41]. A subset of tumours (12/56; 21%) that had undergone genome duplication evidenced asynchronous gains; asynchronous gains were only observed in pure seminomas or NSGCT with a predominant EC or seminoma component, suggesting divergent patterns of chromosomal evolution underlying histogenesis (Fig. 3c). Moreover, the proportion of CNAs attributed to signature CN2 was significantly higher in samples with synchronous gain patterns ($P = 0.029$, Wilcoxon rank sum test), while the proportion of CNAs attributed to CN14 was higher ($P = 0.006$, Wilcoxon rank sum test) in asynchronous genomes.

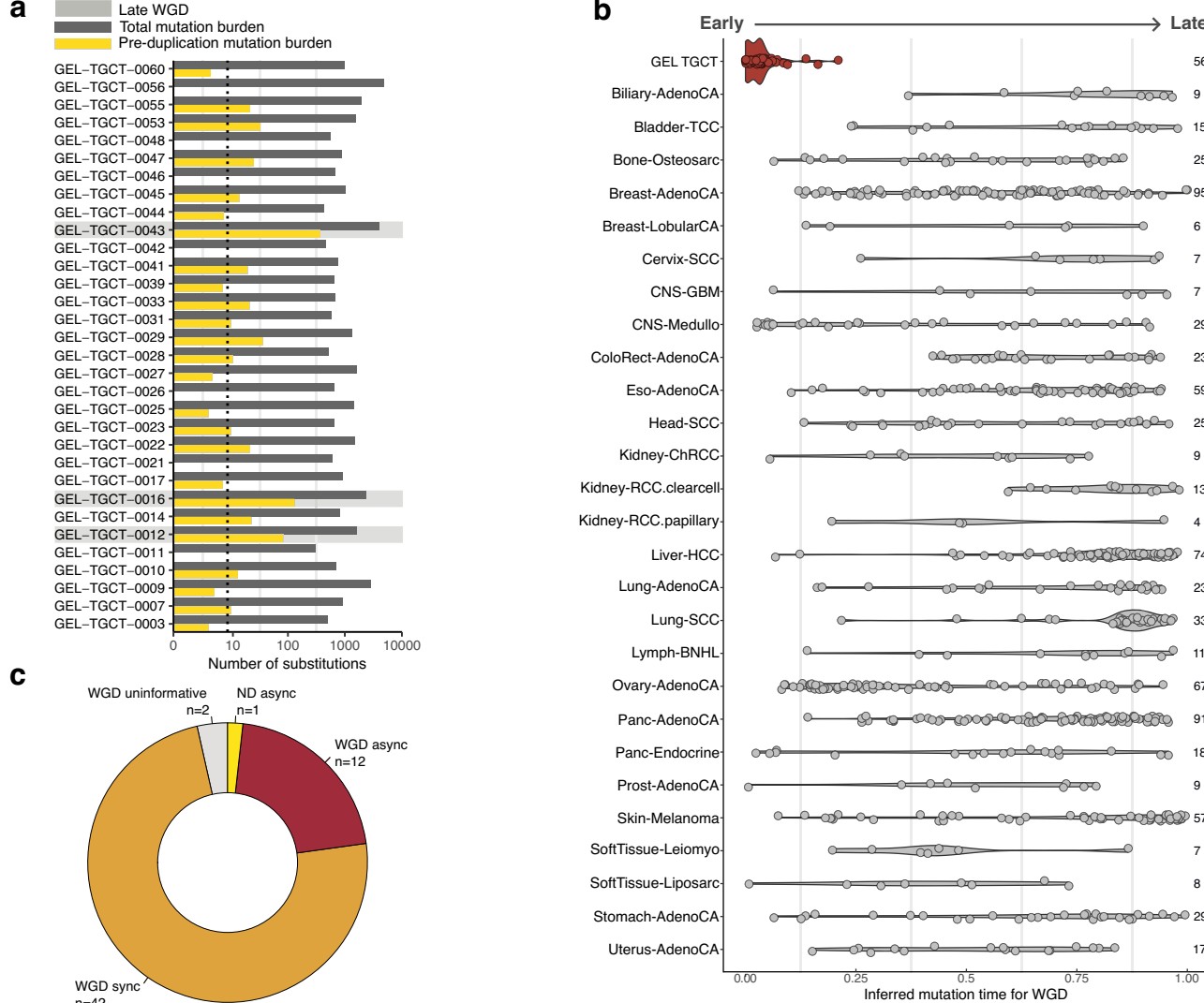

**Fig. 3 | Timing of whole genome duplication (WGD) events across Pan-Cancer Analysis of Whole Genomes (PCAWG) and Genomics England (GEL) testicular germ cell tumour (TGCT) cohorts. a** Bar plot showing estimated pre-duplication mutation burden (yellow) and total clonal mutation burden (dark grey) per TGCT. The dashed line indicates the median pre-duplication burden across all samples. Samples where WGD occurred relatively late are shaded in light grey. **b** Number of samples with a WGD event in each cancer type is shown alongside the corresponding violinplot. Data points from PCAWG appear in grey. Data points from GEL appear in red. PCAWG cancer types with less than four WGD samples not shown. Tumour abbreviations reported as per PCAWG study (ref. 41). **c** Distribution of synchronous (sync) and asynchronous (async) gain patterns across GEL TGCT genomes, split by ploidy status (WGD whole genome duplication, ND near diploid). Uninformative samples had too few mutations or gained segments to allow accurate timing.

## Relative timing of genome doubling and driver mutations in TGCT

Using a permutation approach, we identified CNAs with evidence for significant enrichment or depletion across the GEL TGCT cohort and in the seminoma and NSGCT subgroups (Methods, Supplementary Data 7). A probabilistic timing model was used to reconstruct the order of acquisition of recurrent genomic aberrations, including WGD, enriched CNAs, and putative driver mutations across all TGCT genomes and within each of seminomas and NSGCT (Methods, Fig. 4). Enriched gains spanned known cancer and TGCT drivers including *MYC* (8q11-q24), *EGFR* (7p11.2), and *BRAF* (7q34). Similarly, enriched LOH events covered tumour suppressor genes such as *APC* (5q22.2), *ATM* (11q22.3), and *CDX2* (13q12.2). No evidence was found for enriched homozygous deletion events. We further identified CNAs with evidence for significant negative enrichment in TGCT, implying that these events are less important for, or perhaps incompatible with, driving tumourigenesis in TGCT or in the context of widespread WGD (Supplementary Data 7).

In line with our analysis of WGD developmental timing, tetraploidisation was consistently the earliest event seen, followed by 12p gains spanning the *KRAS* locus, which may imply an initiating tumourigenic role in adult TGCT. To more accurately estimate the timing of high-level copy number gains specific to chromosome 12, we used AmplificationTimeR[44], a method for timing individual amplification events (Supplementary Methods, Supplementary Data 7). Within most samples analysed, findings were in keeping with early timing of whole genome doubling. However, there is evidence to suggest that in some participants, chromosome 12 gains instead represent the earliest occurring events in the evolutionary history of the tumour, occurring pre-WGD (Supplementary Fig. 17).

Most of the early events following WGD were gains and showed balanced representation across TGCTs (Fig. 3b), although later enriched events were specific to TGCT subtypes, such as the 12q11 gain

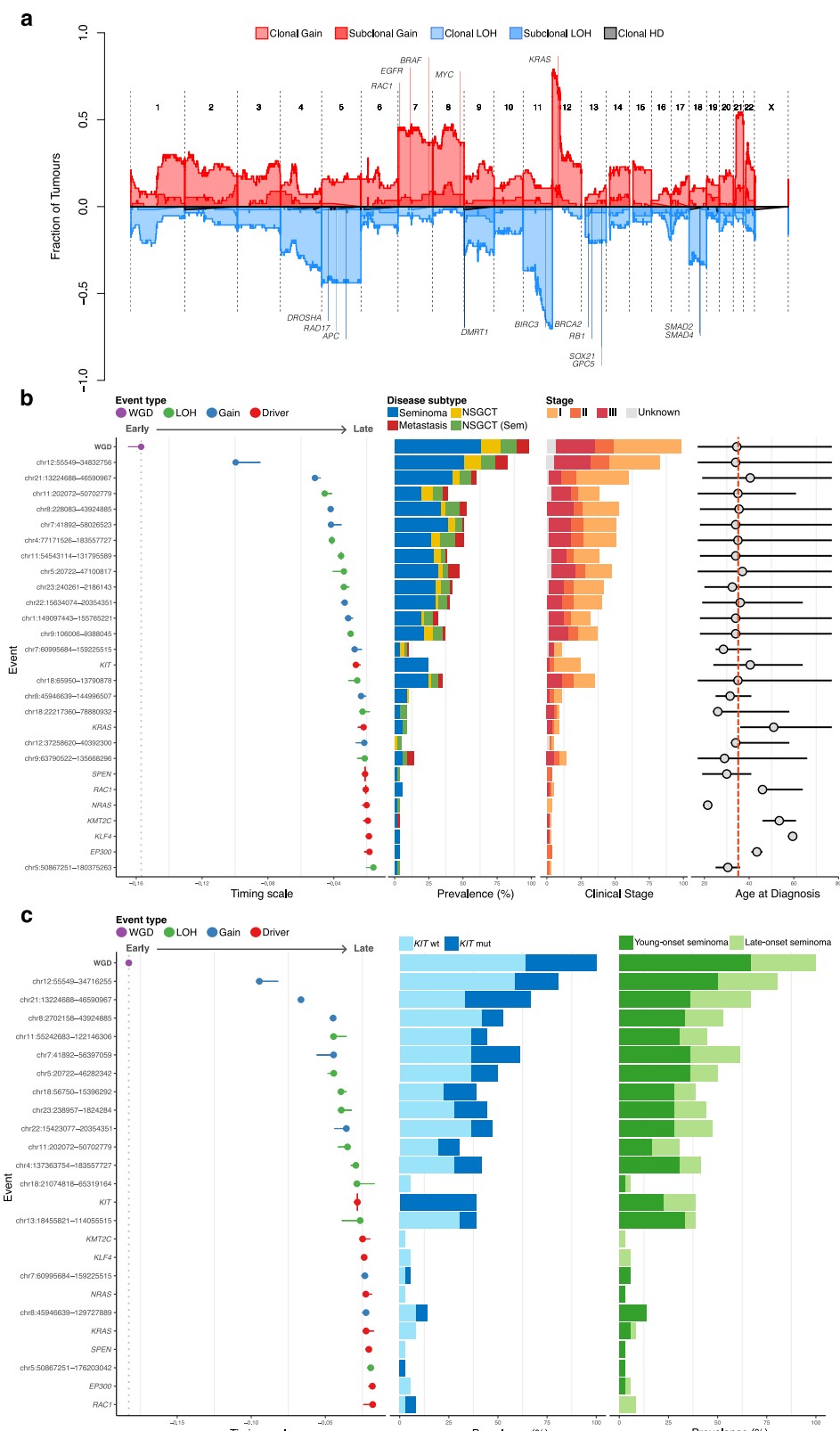

spanning *KIF21A* and restricted to NSGCT. The only CNA event uniquely enriched in seminomas was a recurrent LOH spanning *BRCA2* (chr13:18–114 Mb). Whilst not statistically significant, most individuals (12/14; 86%) harbouring this event belonged to the 'young-onset' group (<40 years; Supplementary Methods). Additionally, a subset of enriched CNAs were specific to young-onset seminomas, such as chr8:45–129 Mb and chr7:60–159 Mb (Fig. 4c).

A Dirichlet Process clustering algorithm was used to cluster SNVs and indels according to their cancer cell fraction[45]. There was no significant variation in the proportion of SNVs, indels, or CNAs identified as subclonal across participants according to tumour stage or subtype (Supplementary Fig. 18a, b; Supplementary Data 7). Multi-site clonality analysis of four whole-genome sequenced regions from one participant (GEL-TGCT-0058; pure seminoma) point towards limited

**Fig. 4 | Probabilistic ordering reveals most likely timing of copy number and driver events in TGCT. a** Genome-wide landscape of clonal and subclonal loss of heterozygosity (LOH), gain, and homozygous deletion (HD) events in Genomics England (GEL) testicular germ cell tumour (TGCT) cohort. The y-axis corresponds to the fraction of tumours with a particular event. Events identified as enriched by our model and genes of interest within these regions are labelled. **b** Probabilistic ordering (left panel) of significantly enriched copy number events, whole genome duplication (WGD), and IntOGen-identified mutational drivers. A Plackett-Luce model was used to order events by sampling from all possible tumour phylogenies across the entire dataset (1000 iterations). Events are ordered along a timing scale (x-axis) from early to late by the mean value of the relative timing estimates. Horizontal lines show the range of time scale values inferred across the cohort for each event. The vertical lines and points for each event represent the mean and standard deviation for each distribution. The grey dashed vertical line represents the mean timing estimate of WGD across all samples. The proportion of these events present in different subtypes and clinical stages is shown in two central panels. Horizontal lines (right panel) indicate the minimum and maximum age of diagnosis in individuals harbouring a mutational event. The shaded grey circle indicates the median age of diagnosis corresponding to each mutational event. The dashed red line shows the median age of the cohort. **c.** Plackett-Luce-based probabilistic ordering of enriched events and mutational drivers only in seminomas. Seminomas were split into two groups (young-onset, late-onset; Supplementary Methods). The grey dashed vertical line represents the mean timing estimate of WGD across all samples. LOH loss of heterozygosity, NSGCT non-seminomatous germ cell tumours excluding tumours with seminomatous components, NSGCT (Sem) NSGCT with seminomatous components.

intra-patient tumour heterogeneity (Supplementary Fig. 18c). Across all subtypes, mutations in driver genes including *KIT*, *KRAS*, and *NRAS* were relatively late events, occurring post-WGD and after corresponding copy number gains or other CNAs (Fig. 4b, c). Participants with *KRAS* and *KMT2C* drivers typically had a higher age at diagnosis.

### HLA loss enriched in seminomas
None of the 60 tumours harboured nonsynonymous mutations in human leucocyte antigen (HLA) genes (Methods). However, LOH at the HLA locus, where either the maternal or paternal allele is lost, was identified in six tumours using the LOHHLA algorithm[46] (Supplementary Data 8). HLA LOH affected a single type-I gene in two seminoma cases, and the HLA-A and -C genes in another three. HLA LOH potentially affected all three HLA genes in a single case, which was the only NSGCT case affected. In GEL-TGCT-0053, a post-chemotherapy lymph node metastasis, LOH was detected in both HLA-A and HLA-B, and although LOH could not be established in this case where two highly similar HLA-C haplotypes were observed (C07:02 and C07:01), the ordering of HLA genes suggested it was likely that there was also loss of HLA-C. We found no significant associations between HLA homozygosity (Methods) and either age of diagnosis, clinical stage, or pathological stage.

Allelic imbalance without LOH, i.e., HLA imbalance as a result of unequal copy gain at the HLA locus or LOH not reaching statistical significance, was observed in a further 17 cases and in both seminomas and NSGCT. The majority of cases where HLA imbalance was observed were in seminomas (9/17; 53%). In most other tumours with HLA imbalance, the major histological component was EC (6/17; 35%), suggesting subtype-specific mechanisms of immune disruption in TGCT. HLA or B2M mutations, which can disrupt neoantigen-MHC binding, were not observed (Methods). Scanning for somatic mutations in genes involved in antigen presentation and processing (Methods), we found one seminoma exhibiting HLA LOH had also acquired a mutation in the proteasome regulator *PSME4*, which plays a key role in immunoproteasome activity and generating immunopeptidome diversity. Collectively, these findings suggest that while HLA mutations are unlikely to be a major mechanism of immune evasion in TGCT overall, HLA LOH could represent a mechanism of immune disruption and/or escape, primarily in a subset of seminomas, though further study is required.

TGCT samples had a median of 10 neoantigenic mutations, mostly arising from SNVs (Supplementary Data 8). We found no significant difference in neoantigen burden between samples with HLA LOH, HLA imbalance, or an intact HLA locus. We next explored tumours with HLA LOH and evaluated whether the LOH event affected their neoantigen landscape (Supplementary Fig. 19). To do this, we computed the number of antigenic peptides predicted to bind the allele lost in the LOH event and compared it with the number of peptides binding to the retained allele. We found no significant difference overall, but observed a trend in 3/6 samples (GEL-TGCT-0007, GEL-TGCT-0018, GEL-TGCT-0050) for a higher number of binders associated with the lost, rather than the retained, allele. In 2/6 samples we observed the opposite trend, though the difference between lost/kept-associated binders was less striking. These observations could suggest that in some seminomas, HLA LOH provides functional escape from immune selection pressure, whereas in the other samples, immune selection is negligible, or HLA LOH is secondary to other non-genetic escape mechanisms.

## Discussion
To our knowledge, this analysis provides the largest study of the whole genome landscape of adult TGCT. Our study extends the number of such whole genomes reported by almost tenfold, further defines the genomic basis of molecular subtypes, and charts the typical evolutionary trajectories of tumours.

Supporting evidence has been provided for 17 candidate driver genes, including subtype-specific drivers. Furthermore, we identified a putative loss-of-function driver mutation in *PTMA*, a gene not currently catalogued by OncoKB or the COSMIC Cancer Gene Census. Previous work has suggested that a homologue of *PTMA*, *PTMS* (parathymosin), may be implicated in GCT epigenetic remodelling[11]. It is likely that putative drivers identified in only one TGCT cohort, such as *KLF4* in GEL or *FAT4* in TCGA, represent rare or low frequency driver events. A recent study of childhood and adolescent GCTs reported that FAT family genes were mutated exclusively in non-seminoma subtypes[9]. Recurrent deletions and structural variant hotspots involving cadherin genes detected in this study support a potentially important role for these proteins in TGCT pathogenesis and progression. Sertoli-germ cell adhesion is crucial for spermatogenesis and cadherin proteins are important mediators of cell-cell adhesion in the testes[47].

We observed a lower burden of SBS1 relative to SBS5 across all participants, which could be due to the age distribution of the cohort[48], with the nominally clocklike SBS1 signature entirely absent in some cases. Recent observation supports the hypothesis that a lower SBS1 burden in normal seminiferous tubules could be due to a reduced rate of spermatogonial stem cell division compared with somatic stem cells, with SBS1 and SBS5 rates of generation independently regulated[49]. It has previously been postulated that SBS1 mutations may be generated during DNA replication, at the time of mitosis[35]. The depleted contribution of SBS1 in *KIT*-mutated seminomas could signal that these cells are, or were, maintained in prolonged mitotic arrest.

In addition, we identified six sporadic SBS signatures. Interestingly, mutational signature SBS32, linked to chronic exposure to azathioprine, was detected in ~10% of participants, without any such recorded treatment history, indicating that other exposures may generate SBS32 mutations. The foetal origins of malignant GCTs raises the possibility that such exposure might even arise *in utero*[50,51]. One further possibility is that exposures at different ages or individual differences in susceptibility to mutational insult could also underlie the bimodal diagnosis age distribution observed in seminomas. The teratoma GCT subtype represents a terminally differentiated tissue which is typically non-responsive to chemotherapy[52]. The marked

absence of signatures SBS31 and SBS35, linked with platinum compound exposure, in post-chemotherapy teratomas reported in our study suggests that this might be due, at least in part, to the ability of non-differentiated cells to withstand the specific mutational damage normally associated with chemotherapy exposure. We report signature SBS18 in adult TGCT, and exclusively in NSGCT, perhaps corresponding to damage from intrinsic ROS mechanisms[53] initially induced during development. Our analysis of mutational signature evolution suggests such processes remain active during the development and progression of NSGCT subtypes. Importantly, a substantial proportion of the signatures detected in TGCT remain of unknown aetiology, highlighting the contribution of mutational processes yet to be identified.

In seminomas, low levels of tumour-infiltrating lymphocytes (TILs) have been linked to poorer patient outcomes, namely a higher clinical stage at presentation and increased relapse rates[54]. Our finding that HLA LOH is almost exclusive to seminomas, and the possibility that a reduced set of antigenic peptides is presented to the immune system as a result[46], suggests a potential genomic mechanism underpinning the low-TILs seminoma subgroup. Additionally, a recent study reported a prevalence of HLA-I LOH of 16.7% in germ cell tumours, which is largely consistent with our results[55].

The chronological ordering of genomic events in TGCT corresponds with the initiation of most tumours within the gonadal developmental pathway. Consistent with the canonical model of TGCT, WGD occurs at the earliest stage, likely arising from erroneous centromeric division during mitotic anaphase, and most often precedes 12p gains[56]. While other fundamental biological abnormalities of TGCT are evident in germ cell neoplasia in situ (GCNIS)[57], whether occurrence of WGD is as frequent in these precursor lesions, or even in normal germ cells devoid of any clinical manifestation, remains to be determined. Probabilistic ordering also supports the idea that chromosomal gains and losses following early tetraploidisation of tumour cells are non-random, with specific events being favoured or suppressed during typical TGCT development. Recent analyses of WGD in ovarian adenocarcinoma suggest that tetraploidisation, though often arising early in clonal evolution, can occur throughout the female reproductive lifespan[41]. However, in male reproductive tissues it appears that such events may be constrained to early life. The less common relatively late-WGD tumours identified here highlight rare aetiologies of TGCT that require further exploration in larger cohorts.

Limitations of the present study include the relatively small sample size and the clinical homogeneity of the cohort, which is enriched for early stage seminomas, leaving small numbers of patients for detailed subgroup analyses. Large targeted studies will add further power to analyse rarer TGCT subtypes, more aggressive forms of disease, and individuals with poorer survival outcomes. A further limitation is that we have only considered a single data modality (DNA). Although we established putative drivers in TGCT pathogenesis, confirmation of our findings in complementary experimental work would increase confidence in their reliability. Analysis of multi-modal data (e.g., RNA, protein, and DNA accessibility) is essential for an improved comprehension of the molecular underpinnings of TGCT initiation and progression. In addition, our analysis did not consider the potential pathogenicity of germline variants. Future studies should address these gaps. Despite these limitations, our study sheds light on the diversity of genomic processes driving TGCT oncogenesis and progression and highlights important genomic alterations that could facilitate immune evasion in specific TGCT subtypes.

## Methods

### Participant recruitment and consent
This study was made possible through access to data in the National Genomic Research Library, which is managed by Genomics England Limited (a wholly owned company of the Department of Health and Social Care). Genomics England has approval from the Health Research Authority Committee East of England – Cambridge South (REC Ref 14/EE/1112). Additional ethical oversight is provided by Genomics England's Ethics Advisory Committee and Participant Panel. Detailed consent for participation in the 100,000 Genomes Project was obtained for all participants in line with cancer-programme specific guidance from Genomics England (https://files.genomicsengland.co.uk/forms/Cancer-Model-3.2.0.docx). Participants recruited to the study included anyone with a likely diagnosis of a testicular tumour, regardless of sex or gender, in accordance with SAGER guidelines. The sex of participants was reported by submitting clinical teams and was not determined by the research team.

### Tissue collection and handling
In the 100,000 Genome Project, tissue is collected then processed at UK National Health Service (NHS) Genomic Medicine Centre hubs (GMCs). Blood germline and fresh frozen tumour biopsy samples were collected and processed according to the specifications outlined in the Genomics England Sample Handling Guidance Documentation Version 4.0 (https://files.genomicsengland.co.uk/forms/Sample-Handling-Guidance-v4.0.pdf). Formalin-fixed paraffin-embedded tumour samples were excluded from the current study.

### Sample selection
Sixty whole genome sequenced tumours were analysed in this study, including a set of four primary tumour regions sampled from a single individual. For the identification of driver mutations, recurrent copy number alterations, and amplicons, we used 54 out of the 57 individual samples. Specifically, we selected the highest purity tumour from the four multi-region samples (see Supplementary Data 1). Additionally, we excluded genomes that were not generated using PCR-free library preparation (2 out of 57), as well as one metastatic case that had undergone malignant somatic transformation (GEL-TGCT-0056; see Supplementary Data 1). For more details, please refer to Supplementary Information.

### Clinical data
Clinical data was collected from NHS GMCs via the central Genomics England team in line with Genomics England Cancer Model data submission and sample tracking guidance.

Version 3.2.0 (https://files.genomicsengland.co.uk/forms/Cancer-Model-Sample-Tracking-3.2.0.docx).

### Variant calling and filtering
Detection of germline and somatic single nucleotide variants (SNV) and insertions/deletions (indels) < 50 bp was performed using Strelka[58] (version 2.9.9). Alongside default Strelka filters, we applied the following additional filters to remove variants including[59]:
- Variants with population germline allele frequency ≥1% in the 100kGP or gnomAD datasets.
- Variants with excessive somatic frequency (≥5%) in the 100kGP cancer dataset. The 5% threshold was based on the frequency of recurrent non-synonymous variants in hallmark genes in the Cancer Gene Census[60].
- Variants identified as simple repeats by Tandem Repeats Finder[61].
- Indels where ≥10% of base calls within a 50-base window on either side of the indel were flagged and filtered by Strelka due to high sequencing noise.
- Variants called in regions of poor mappability where the majority of overlapping 150 bp reads do not map uniquely to the variant position.
- Variants resulting from systematic mapping and calling artefacts present in both tumour and normal 100kGP sample sets. Specifically, somatic SNVs were identified where the ratio of tumour allele depths differed significantly from the ratio of allele depths

at the same site in a panel of normal samples (PoN), as tested using Fisher's exact test. The PoN comprised 7000 non-tumour genomes from the GEL dataset. Only individuals not carrying the relevant alternate allele at a particular site were used to count allele depths in the PoN. To ensure similarity to the Strelka preset filters, duplicate reads were removed and quality thresholds set at base quality ≥ 5, mapping quality ≥ 5, and phred score < 80.

SNVs and small indels were normalized (left aligned, trimmed, multi-allelic variants decomposed) and annotated using Cellbase with GRCh38 Ensembl v90, COSMIC[60] (version v86/GRCh38) and ClinVar[62] (October 2018 release) databases. Variant consequences were annotated using a high-performance tool within Cellbase, and only variants associated with a set of curated consequence types (e.g., stop gained/lost, start lost, frameshift, inframe insertion/deletion, missense, splice acceptor/donor, and splice region variants) in canonical transcripts were reported.

### Driver identification

Cancer driver genes were identified using the IntOGen pipeline[63]. The relative evolutionary timings of candidate driver mutations were obtained using MutationTimeR[41].

### CNA analysis

A Nextflow pipeline, in combination with the CleanCNA R package (https://github.com/afrangou/CleanCNA), performed the following steps. Battenberg v.2.2.7 (https://github.com/Wedge-lab/battenberg) was run across all samples and the resulting genome-wide copy number profiles were then input into two algorithms, DPClust[45] (version 2.2.5; https://github.com/Wedge-lab/dpclust) and CNAqc (https://github.com/caravagnalab/CNAqc), in conjunction with Variant Allele Frequency (VAF) information from somatic SNVs. DPClust was used to calculate the Cancer Cell Fraction (CCF) of individual mutations, and cluster mutations based on their CCF. CNAqc compared the expected peaks of SNV VAF distributions with the observed peaks for an individual sample, in a set of 5 copy number states [1:0,1:1,2:0,2:1,2:2]. Metrics from the Battenberg profile, DPClust output, and CNAqc output, were combined in order to provide an overall assessment of the CNV profile. Extrachromosomal DNA (ecDNA) molecules were detected from tumour BAM files using Amplicon Architect (v1.2)[33]. Recurrent arm-level copy number events, as well as focal amplifications and deletions, were identified using the GISTIC[24] algorithm (v2.0.2.3; https://github.com/broadinstitute/gistic2).

### Classifying whole genome duplication events

We defined samples that had undergone WGD based on tumour ploidy and the extent of loss of heterozygosity. Individual tumours (including only the highest tumour purity sample from multi-region sampling) were plotted based on average ploidy and fraction of genome with LOH (Supplementary Fig. 16). The separating line between WGD and non-WGD tumours was estimated according to the approach established by Dentro et al.[64] as $y = 2.9 - 2x$. Most samples with WGD evidenced synchronous chromosomal gains[41], further validating this approach. To evaluate this, we used the same approach taken by the PCAWG Consortium[64].

### Structural variant (SV) analysis

Somatic rearrangements were identified using a graph-based consensus approach comprising Delly[65] (version 0.7.8), Lumpy[66] (version 0.2.13), and Manta[67] (version 0.28.0), whilst also considering support from CNAs. Rearrangements were first called using the three individual callers with default parameters. Delly was run with post-filtering of somatic SVs using all normal samples. Rearrangements from the three individual callers were further filtered if any reads supporting the variant were identified in the matched normal, if < 2% of tumour reads

supported the variant, or if either variant breakpoint was located in a telomeric or centromeric region or on a non-standard reference contig (i.e. not chromosomes 1–22, X or Y). Remaining rearrangements were merged with a modified version of the PCAWG SV Merge tool, which uses a graph-based approach to identify and merge rearrangements identified by multiple callers, allowing 400 bp slop for breakpoint positions[68]. Rearrangements were included in the final data set if they were identified by at least two callers, or by a single caller but with a breakpoint within 3 kb of a CNA segment boundary.

### Mutational signatures analysis

Mutational signatures were extracted from SBS, DBS, ID, copy number alterations and structural variants using SigProfilerExtractor[69] (version 1.1.3). All signature extraction runs were performed using random initialization (*nmf_init* parameter), 500 NMF replicate runs (*nmf_replicates* parameter) with 10,000 (*min_nmf_iterations* parameter) to 1,000,000 NMF iterations (*max_nmf_iterations* parameter). For SBS signatures, we assumed the presence of 1–30 signatures, for DBS signatures we assumed 1–20, for ID signatures 1–15, for CN signatures 1–30, and for SV signatures 1–15 signatures. In downstream analysis, only DBS signatures with > 4 mutations were retained. The optimal number of de novo signatures was determined by aiming to maximize the mean sample cosine distance while ensuring that the average stability remained above 0.9.

Linear regression models were fit to test the association between patients' age and the number of mutations attributed to each signature. Assignment of SNVs to SBS mutational signatures was carried out for all mutations from each sample. Briefly, trinucleotide contexts were obtained for all SNVs processed by DPClust, where each SNV has been assigned to a cluster with CCF values allowing for clonal tree reconstruction. For clonal and subclonal clusters, probabilities per signature were assigned by multiplying the proportion of mutations of a specific mutation type assigned to that cluster against the corresponding decomposed mutational probability assigned by SigProfilerExtractor, and summing across these values.

Kruskal-Wallis tests were used to compare structural variant signature activities among samples, grouped by tumour type (primary versus metastatic), broad disease subtypes, and detailed disease subtypes (see Supplementary Data 1).

### HLA LOH calling with LOHHLA

Somatic mutations in the HLA locus were predicted using Polysolver[70]. Loss of heterozygosity at the HLA locus was predicted using LOHHLA[46]. A type-I allele of a sample was annotated as "allelic imbalance" (AI) if the *p*-value testing the difference in evidence for the two alleles was < 0.01. Alleles with AI were further labelled as LOH if the following criteria held: (i) the predicted copy number of the lost allele was below 0.5 with confidence interval strictly below 0.7; (ii) the copy number of the kept allele was above 0.75; (iii) the number of mismatched sites between alleles was above 10. Samples with at least one HLA gene showing LOH were labelled as "HLA LOH", and samples with no LOH but at least one gene showing AI as "HLA imbalance". Note that HLA imbalance could be a result of unequal copy gain in the HLA locus (e.g. CN = 2:1), but could also indicate LOH that does not reach statistical significance, e.g. because of low sample purity or because it is subclonal in the tumour.

### Reporting summary

Further information on research design is available in the Nature Portfolio Reporting Summary linked to this article.

## Data availability

Summary information for each sample is provided in the Supplementary Data, ensuring that data do not enable the identification of participants. Primary data from the 100,000 Genomes Project are held

in a secure Research Environment and are available to registered users. To become a member of the Genomics England research network and obtain access, please visit https://www.genomicsengland.co.uk/research/academic/join-gecip. The process involves an online application, verification by the applicant's institution, completion of a short information governance training course, and final approval by Genomics England. More information is available at https://www.genomicsengland.co.uk/research/academic. The Genomics England data access agreement can be found at https://doi.org/10.6084/m9.figshare.4530893.v7 (ref. 71). WGS data and processed files from this project can be accessed by joining the Pan-Cancer and Molecular Oncology community, after data access has been approved (https://www.genomicsengland.co.uk/research/pan-cancer-and-molecular-oncology-community). All analyses of Genomics England data must take place within the Genomics England Research Environment (https://re-docs.genomicsengland.co.uk/). The 100,000 Genomes Project publication policies can be found at https://files.genomicsengland.co.uk/images/Publication-Policy-v5.0.pdf. Please see https://re-docs.genomicsengland.co.uk/data_overview/ for further information. The TCGA TGCT genomic dataset reanalysed here is available from cBioPortal via https://www.cbioportal.org/study/summary?id=tgct_tcga_pan_can_atlas_2018 (ref. 5). A SFTP server is available to access the PCAWG data referenced in our study. More information on accessing this data can be found at https://docs.icgc-argo.org/docs/data-access/icgc-25k-data#accessing-icgc-25k-release-data.

## Code availability

The code for the WGS subclonal copy number caller can be found at https://github.com/Wedge-lab/battenberg (v.2.2.7). The code for the DPClust R package used for tumour subclonal reconstruction can be found at https://github.com/Wedge-lab/dpclust (v.2.2.5). The code for inferring the order of genomic events can be found at https://github.com/hturner/PlackettLuce. The code for the chronological timing analysis can be found at https://gerstung-lab.github.io/PCAWG-11/ and https://github.com/gerstung-lab/MutationTimeR. The code used for filtering CNAs can be found at https://github.com/afrangou/CleanCNA. The code used to identify rearrangement hotspots can be found at https://github.com/DominikGlodzik/hotspots/tree/glodzik2016/. The code used to time amplifications relative to WGD can be found at https://github.com/Wedge-lab/AmplificationTimeR.

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

## Acknowledgements

We thank the participants for contributing to this study. This research was made possible through access to the data and findings generated by the 100,000 Genomes Project. The 100,000 Genomes Project is managed by Genomics England Limited (GEL; a wholly owned company of the Department of Health and Social Care). The 100,000 Genomes Project is funded by the National Institute for Health Research (NIHR) and NHS England. The Wellcome Trust, Cancer Research UK, and the Medical Research Council have also funded research infrastructure. The 100,000 Genomes Project uses data provided by participants and collected by the National Health Service as part of their care and support. This work was co-funded by the NIHR Manchester Biomedical Research Centre (NIHR203308). C.V. is partly funded by the NIHR Oxford Biomedical Research Centre (BRC). The views expressed are those of the author(s) and not necessarily those of the NHS, the NIHR, or the Department of Health and Social Care. A.J.C., D.C., B.K., and R.S.H. were supported by funding from the Wellcome Trust (214388) and Cancer Research UK (C1298/A8362). The authors would like to acknowledge Dr Sam Behjati and Thomas Oliver of the Wellcome Sanger Institute for interesting discussions and Peter O'Donovan for technical support.

## Author contributions

C.V., M.J.M., A.P.., and D.C.W. planned the study. V.W. acted as a participant representative for the GECIP. R.Y.B., R.S., K.B., D.S., R.J.Y., A.T.J.L.,

coordinated participant recruitment and sampling at GMCs. M.N.L., A.F., B.K., A.J.C., D.C., A.J.G., P.A., P.L., I.P., G.M.J., A. Sahli, A. Sosinsky, A.T., E.M.S., and E.L. contributed to data processing and handling, quality control, and analysis. R.S.H. supervised parts of the analysis. M.N.L. wrote the manuscript with input from all co-authors.

## Competing interests

Genomics England is a company wholly owned by the UK Department of Health and Social Care and was created in 2013 to introduce WGS into healthcare in conjunction with NHS England. Authors affiliated with Genomics England are, or were, salaried by or seconded to Genomics England (A. Sosinsky, P.A.) or act as a GeCIP-specific patient/participant representative (V.W.). All other authors declare they have no known competing financial interests or personal relationships that could have appeared to influence the work reported in this paper.

## Additional information

[1]Big Data Institute, Nuffield Department of Medicine, University of Oxford, Oxford, UK. [2]Ludwig Institute for Cancer Research, Nuffield Department of Medicine, University of Oxford, Oxford, UK. [3]Department of Microbiology, Moyne Institute of Preventive Medicine, School of Genetics and Microbiology, Trinity College Dublin, Dublin, Ireland. [4]Max Planck Institute of Molecular Cell Biology and Genetics, Dresden, Germany. [5]Division of Genetics and Epidemiology, The Institute of Cancer Research, London, UK. [6]University College London Cancer Institute, 72 Huntley Street, London, UK. [7]Department of Mathematical Sciences, Chalmers University of Technology and University of Gothenburg, Gothenburg, Sweden. [8]Genomics England, London, UK. [9]Department of Biology, University of Konstanz, Universitaetsstrasse 10, D-78464 Konstanz, Germany. [10]Manchester Cancer Research Centre, The University of Manchester, Manchester, UK. [11]Division of Cancer Sciences, University of Manchester, Manchester Academic Health Science Centre, Manchester, UK. [12]Christie Hospital, The Christie NHS Foundation Trust, Manchester Academic Health Science Centre, Manchester, UK. [13]Norfolk and Norwich University Hospitals NHS Foundation Trust, Norwich, UK. [14]Guy's and St Thomas' NHS Foundation Trust, London, UK. [15]Department of Pathology, The Royal Wolverhampton NHS Trust, Wolverhampton, UK. [16]Leeds Institute of Medical Research at St James's, University of Leeds, Leeds, UK. [17]Weston Park Cancer Centre, Sheffield Teaching Hospitals NHS Foundation Trust, Sheffield, UK. [18]Department of Oncology, Oxford University Hospitals NHS Foundation Trust, Oxford, UK. [19]Department of Paediatric Haematology and Oncology, Cambridge University Hospitals NHS Foundation Trust, Cambridge, UK. [20]Department of Pathology, University of Cambridge, Cambridge, UK. [21]NIHR Oxford Biomedical Research Centre, Oxford, UK. [22]Nuffield Department of Surgical Sciences, University of Oxford, Oxford, UK. [23]These authors contributed equally: Anna Frangou, Ben Kinnersley. [24]These authors jointly supervised this work: Andrew Protheroe, Matthew J. Murray, David C. Wedge, Clare Verrill. ✉e-mail: mnileathlobhair@gmail.com; mjm16@cam.ac.uk; david.wedge@manchester.ac.uk; clare.verrill@ouh.nhs.uk

## Testicular Cancer Genomics England Clinical Interpretation Partnership Consortium

Richard Y. Ball [13], Ksenija Benes[15], Daniel Chubb[5], Alex J. Cornish [5], Anna Frangou [4,23], Ben Kinnersley [5,6,23], Alexander T. J. Lee[12], Matthew J. Murray [19,20,24]✉, Máire Ní Leathlobhair [1,2,3]✉, Andrew Protheroe [18,24], Rushan Sylva[14], Dan Stark[16], Clare Verrill [21,22,24]✉, David C. Wedge [1,11,12,24]✉ & Robin J. Young[17]

## Genomics England Research Consortium

Alona Sosinsky [8]

A full list of members and their affiliations appears in the Supplementary Information.

