## [Peer Review File · Nature Communications]

REVIEWER COMMENTS

Reviewer #1 (Remarks to the Author): Expert in TGCT genetics

Please see the attached file.

Reviewer #2 (Remarks to the Author): Expert in cancer genomics and evolution

Lethlobhair et colleagues present a thorough analysis of whole genome sequencing data of from testicular germ cell tumors (TGCT). While not an exceedingly large study by today's standards (60 tumors), and smaller than the TCGA study on TGCT (~130), it is nevertheless the most comprehensive WGS analysis of TGCT thus far. While there are no real breakthrough insights into driver mechanisms or similar - despite a few proposed new driver genes/mutations - the analyses appear generally to be carried out in an expertly manner and the results should constitute a useful contribution to this field of research.

Specific comments:

Regarding newly identified drivers (KLF4, RAC1, EP300) - how come these were not found e.g. in the larger TCGA cohort? This is worth discussing. Are they mutated to some extent without reaching significance?

Intuitively, I am somewhat sceptical that negative selection in ncDNA can be reliably detected in such a small low-burden cohort (as signals of negative selection have proved very limited even in coding DNA and even in large cohorts). I'm not saying these results are necessarily wrong, but selection is after all difficult to assess - so I think the authors should consider using a more cautious statement (now: "is likely to"). Also in discussion "evidence" for this is mentioned, which I personally think is a too-strong word: it is impossible to gain "evidence" of selection in these analyses: a low P value simply means there is a deviation the modelled expectation and the model is always wrong to some extent (the question is just how much).

I believe WGD and diploid are incorrectly labelled in Fig. S15. It is not obvious how this classification was carried out and information in the Fig. S15 is limited, although there are a few notes in the Methods supplement about a regression line by Dentro et al. It's hard to judge how reliable this classification is. Minimum, perhaps some clarification can be added to the legend to provide some general guidance about how this was done.

I don't see how Fig. 4b supports the claim about synchronous chromosomal gains? Please clarify.

In presenting the mutational timing results, which are rather complex, perhaps the limitations (and underlying principles) can be shortly mentioned in just a sentence or so - e.g. limitations regarding

deletions and mutations in non-CNA or deleted regions.

Minor:

The supplement is weirdly formatted with blank pages and legends often appearing on the next page - or being split between two pages. Also, some of the pictures are not readable due to pixelated text e.g Fig. S8.

Reviewer #3 (Remarks to the Author): Expert in TGCT genetics and genomics

This manuscript comes from Genome England and describes the whole genome landscape of TGCT. This manuscript was incredibly frustrating to read, as the Figures and Supplemental Figures are poorly labeled, they appear to have cut and pasted sections from other papers in the Methods section that do not apply here, and multiple other errors were made (i.e. Figure 5 comes before Figure 4 in text, and Figures 5b, 5c, and 5d are never referred to). To note, it is not be clear that the origin of TGCT is the primordial germ cell, as recent studies have suggested it could be the gonocytes, so the authors need to adjust their language about the cells of origin and not be quite so specific – e.g. the term fetal germ cells, which covers both possibilities.

In general, the manuscript presents interesting whole genome data that explores the mutational signatures and timing of events during tumor development. However, much of what is shown is not entirely novel, such as the timing of KIT mutations after WGD, but the figure showing the timing is elegant and of interest. They also present a deep dive into mutational signatures and demonstrate HLA loss, which provide novel data in the field. However, they have several essentially anecdotal points in the results, which could easily be removed, so that they can focus on novel elements of their data. They also make a number of off hand comments that are not clearly supported by the data.

1. It is not quite correct to say a comprehensive description of the genomic landscape of TGCT is lacking. Beyond the TCGA on TGCT (PMID: 29898407), an overlapping author group published an in-depth examination of TGCT in 2022 (PMID: 35953478). The latter paper also included whole genome sequencing data, although the focus of the analysis was somewhat different.

2. The bimodal peak with the secondary peak occurring in the late 50s, does not seem consistent with what one would consider classic TGCT. Is this epidemiology reflective of the distribution of TGCT diagnosis in the UK?

3. Table S3 is labeled as 'Table S3. List of all substitutions and indels called across the GEL TGCT cohort.' To note, please fix the typo. The Table is 1239 lines long and 1234 of the variants occur just once, so it can't include all substitutions and indels, as described in the text.

4. Should Table S5 precede TableS4, as it is the data from the GEL cohort, and Table S5 contains the

combined cohorts. What the column CGC_Cancer_Gene, which is entirely FALSE in both tables indicates is not clear, since many of the genes are cancer genes.

5. To note that their analysis did not show significance in mutual exclusivity between KIT and KRAS/NRAS, so although true, not sure of the meaning of their analysis in Supp Fig 3. Additionally, they mention that KIT mutations are clustered in intracranial TGCT, but not in classic TGCT?

6. Figure 2 – Given that the Figure 2 includes the commonly amplified and deleted genes at the top, which are not referred to until later in the manuscript, it is confusing for the reader. Perhaps the sections of text can be reorganized so that the reader isn't trying to parse through trying to follow the Figure well before that section of text.

The commonly amplified and deleted genes listed in the figure are not discussed in the text – e.g. BCOR, SMAD4, MEN1 and CDH17. Additional information needs to be provided about the genes that they are highlighting in Figure 2. Presumably they just select genes from their GISTIC results to include in Figure 2 based on their description in 8.3.3 (Methods), but it is not stated. Usually the copy number analysis prior to running GISTIC is used to determine most commonly deleted/amplified genes (in this case Battenberg per-tumor segmentation output, which needs to be referenced). Were cut-offs for deletion and amplification used for this analysis.

Their criteria for calling a missense significant is not clear – e.g. what criteria did the two missense variants in EP300 meet (a gene in which you expect loss of function mutations) to be included as candidate driver mutations? What is the double color (for KIT mutations mainly, missense/amplified) – needs to be clear. What does multi-hit mean?

7. Table S8 – Their analysis suggests that 27% and 18% of non-seminomas in the TGCT TGCA analysis have KIT and KRAS mutations, which is not the case (see Figure 2 from the manuscript and below). It's pretty clear that KIT/KRAS mutations are associated with seminomas. Maybe they have entirely mislabeled their Table – should it be seminomas? They also identify KIT mutations in several of the NSGCT in their own cohort (see Figure 2), again as this phenomenon is not reported in other cohorts, they need to clarify – are they mixed/combined tumors.

8. Hard to know what to make of their comment about FAT4, since it does not show up at all in their cohort.

10. Page 8, middle - It is not at all clear how the authors can state that 'functional integrity of non-coding elements is likely to play a role in TGCT oncogenesis'. Based on what data? They show one significant CTCF site, and a few others that they have picked up – is that more than they would see based on the number of analyses that they have done by chance alone? Certainly it's not enough to support the statement they make.

11. Why 12p not show up at all in their GISTIC analyses? For the most common amplification in TGCT not being identified in their GISTIC analysis seems quite odd. Similarly, why did 11p not show up as part of the focal amplification analysis? Section 8.5 of the methods suggests that they did look for it.

12. Most of the Supplemental Tables have undefined columns. For example, what do d.seg, rate.factor, d.bg, d.obs.exp and RD mean in Table S13. A key to the different column name should be provided throughout the Tables.

13. The discussion about the single tumor that has a somatic-type malignancy (Pages 9-10) does not add much to the manuscript and can be easily removed.

14. In the mutational signature, there are several sets of data quoted not shown, including the differences by KIT mutation status, GEL-TGCT-0038, all of the other tumors about which there is specific discussion and supporting statement such as ID6 and ID8 are mutually exclusive. The point of this section is not clear, as it is just anecdotally discussing specific tumors (without showing their signatures). It is important that they show the SBS signature breakdown by tumor, especially as they are discussing various signatures as potentially of interest – e.g. SBS32. As that signature is seen in a very small proportion overall, whether or not it is relevant is unclear.

What are they trying to show in Supplementary Figure 11, and how does it relate to the main text? It's not mentioned anywhere. A large component of Supplementary Figure 11 is not labeled (below the Figure Legend). What do the headers on the figures mean? It would be helpful if they provide more explanation (as in the Nature source paper).

15. It is very difficult to follow the discussion about the copy number signatures as the supporting data are not presented (or maybe it's in the uninterpretable Supplementary Figure 11).

16. The concluding sentence on Page 12 does not entirely make sense. Two signatures are referenced that are associated with TP53 mutations and BRCA2 mutations, neither of which are found in TGCT. Lots of detail are provide in the methods about extracting structural variants, although it is not clear how much was actually included in the manuscript – e.g. sections 9.4 – SV hotspots – 9.4.1 - Relationship between genomic features and SV rates – 9.4.2 - Permuting SVs - 9.4.3 Identifying SV hotspots – where are these data presented?

17. Supplementary Figure 15 – is the labeling of WGS vs Diploid reversed, as the Figure suggests only one sample has WGD.

18. The text skips from Figure 3 (page 11) to Figure 5 (page 13) – did the authors reverse Figure 4 (Page 15) and 5 in the submission? Why does it say Figure 4b at the end of page 13 – they mean Figure 5b.... No reference to Fig. 5c, 5d are in the text. What are they trying to show with these Figures?

19. The section mid-13 again discussing specific cases in a relatively anecdotal fashion, does not seem to contribute much to the manuscript.

20. Why is the isochromosome section different than the focal amplification section? Why do they not include any supporting data? Supplementary Table 12 refers to the GISTIC data, which as noted above does not show any i12p? The authors state that their finding contradicts prior studies that used i12p

diagnostically for TGCT- the reference they use is from 1989. It is not used currently in that way. Do they find any target genes when they are evaluating the i12p? What genes are amplified is of interest, given the on-going discussion about the target genes of i12p.

21. In Supplemental Figure 16, what do the abbreviations mean? They need to be detailed in the Figure legend.

22. How did the authors come up with 17 candidate driver genes? Table 2 would suggest 16, if you include the genes without any mutations in this sample set (which seems improbable). What evidence is there that PTMA could be a target for therapeutic intervention? (Just seems a throwaway sentence) It's an immune function gene.

23. Finding SBS18 in one sample is not clearly indicative of an important event.

Minor Comments

1. A few grammatical notes – 'While' is a measure of time, and should be used as such. 'Although', 'however' and 'whereas' can be used instead. 'These', 'this' or 'there' should always be defined by a noun.

2. Page 6, end – Sentence starting 'However,.....' doesn't follow as however, doesn't make sense.

3. Supplementary Figure 8 is not readable.

4. Top of Page 8 – Sentence starting 'We assessed.....'. It is not clear that the authors are generally referring their findings (17% of what denominator) and not PTMA specifically, so the sentence needs to be clarified.

5. Supplemental Figure 9. What is the difference between the left and right columns? Some of the signatures are repeated so it's not clear at all. Additionally, what do the numbers below the gray/white blocks mean? (not explained in figure legend)

6. Figure 3c is referred to in the text, and suggests a correlation with age, which is not shown in Figure 3b (no Figure 3c).

7. Page 15 – 'Participants...' missing with.

8. Page 15 – What does 'young-onset older-onset' mean?

Reviewer #4 (Remarks to the Author): Expert in cancer genomics and evolution, and HLA-LOH

This is an interesting paper with rigorous analysis of WGS data, including mutational signatures, CNAs, predicted mutation time

Validation of previous known features of TGCT. Molecular timing results showing early whole genome duplication but late driver mutation events. I think this paper overall is more valuable for the genomic data it has generated rather than the results, which do not tell as compelling of a story.

Comments:

- Biggest caveat: missing RNA-seq, so impossible to say what these genomic differences correspond to phenotypically

Could maybe reanalyze the TCGA RNA-seq data in light of their new results?

- Should have included the 8 other samples from the previous largest WGS study of TGCT mentioned in the paper: <https://doi.org/10.1038/s41467-022-31375-4> (both Wedge and Murray are co-authors, and data is on EGA)

- Too small/heterogeneous of a cohort, hard to make any significant comparisons

- More in depth analysis of differences according to clinical features (age, treatment response, survival if available?)

Bimodal distribution of age of onset is interesting; a figure highlighting the differences between two groups would be more compelling

- Could perform neoantigen pMHC binding affinity prediction, and compare that to HLA LOH

- No p-values in any of the plots, most notably 3b. Also in 3b, the boxplots are covering the data points, order should be switched

- Said they extracted 4 tumor samples from the same patient, were those sequenced separately? If so, should look into heterogeneity of the tumor

- I'm curious if germline HLA homozygosity is also associated with the risk of these tumors and perhaps the age of diagnosis.

- It would be helpful for the readers if the authors can provide a summarizing diagram highlighting their main findings for each subtype.

Overall, this paper provides a useful overview of the general whole genome landscape of this understudied cancer, but doesn't go into much depth with regards to the biological/clinical relevance (maybe due to the limitations of the data collected).

This manuscript reports outcomes of a comprehensive WGS study on adult TGCT cases administered in seven centers located within the UK. Although the scientific contents described in this manuscript may be worth attention from investigators in the relevant fields of research, multiple issues prevent me from recommending the current manuscript for publication.

I must start with an unusual comment – namely, the manuscript is organized very poorly. There are errors in the text that refer to wrong figures or supplementary figures. There are figures or supplementary figures that are not mentioned in the text even once. In one figure, critical symbols are flipped. These errors are too significant that affect my assigned task to examine the scientific merit thoroughly and fairly. Shown below is a list of such errors that I caught, but I am not sure whether my list is exhaustive, or some errors may have escaped my eyes.

- 1) Authors state, "... however only SBS5 and the number of C>T mutations at NpCpG trinucleotides correlated with age ($p = 4.3 \times 10^{-8}$ and $p = 0.02$, respectively; Fig. 3c)." There is no panel C in Figure 3.
- 2) Supplementary Figure 2 is never CORRECTLY mentioned in the text. There is a WRONG mention to Supplementary Figure 2 in the context, "Mutual exclusivity analysis revealed ... FLI1, MAF, and SMAD4 (Supplementary Fig. 2)," but this should refer to Supplementary Figure 3.
- 3) "This approach revealed two structural variant (SV) signatures (S1, S2; Supplementary Figs. 14-15), present in both seminomas and NSGCTs." This appears to be incorrect and should be Figs. 13-14.
- 4) "In all but one case (56/57), tumours were shown to have undergone WGD (Supplementary Fig. 15)." Supplementary Figure 15 shows only one tumor with WGD (red circle) whereas all other datum points shown with blue circle symbol are Diploid
- 5) "This is in stark contrast to most solid cancers, where WGD events are broadly distributed throughout clonal evolution and likely stochastic (Fig. 5a)." Authors skipped any mention to Figure 4 before referring to Figure 5 here. The major problem here is that Figures 5b, 5c, or 5d are never mentioned in the text.
- 6) "A probabilistic timing model was used to reconstruct the order of acquisition of recurrent genomic aberrations, including WGD, enriched CNAs, and putative driver mutations across all TGCTs and within each of seminomas and NSGCTs (Methods, Fig. 3b,c)." This sentence should refer to Figures 4b and 4c.
- 7) Supplementary Figure 17 is never mentioned in the text.

The following issues may not be errors, but Authors are requested to explain and/or make adequate corrections:

- 1) The text describes that 60 tumor samples were analyzed, but Figure 1b presents only 58 tumors. The figure correctly present 39 primary seminomas, but it shows only 15 primary NSGCTs although it should be 16. The figure shows only 4 metastatic tumors, but the text refers to 5 metastatic tumors.
- 2) Author states, "Eight genes were significantly somatically mutated," but these genes are not listed. I had to take a look at the supplementary tables to figure out these eight genes. Please list these genes in the text.
- 3) Authors state, "Eight genes were significantly somatically mutated, including KIT (24%) and its downstream mediators KRAS (7%) and NRAS (4%) (Fig. 1b). But Figure 1b only shows KIT mutations. Authors are requested to add KRAS and NRAS mutation profiles on Figure 1b.
- 4) Authors may want to consider substituting the subtitle "Focal amplifications" with "Extrachromosomal DNA."
- 5) "Though not observed in NSGCTs within the GEL cohort with a predominant YST component, both cases with SBS18 harboured a YST component." Please specify which cases mentioned here as "both cases." The preceding sentences refers to one case, but it is not easy to identify the other case mentioned here.
- 6) Supplementary Figure 9 shows two pictures in each panel. I am assuming that the right panel is for nonclonal mutations and the left is clonal mutations. The legend should clarify this matter.
- 7) "... and associated with TP53 drivers (RefSig.R2) and unclustered deletions up to 100kb associated with BRCA2 mutations (RefSig.R5), respectively (Ref# 32)." The titles shown in Supplementary Figure 14 are truncated (right columns) and cannot be verified for their links to TP53 or BRCA2.
- 8) "We observed a median of just ~9 substitutions (range 0-375) occurring prior to WGD, and in seven cases we did not observe any pre-WGD substitutions, supporting early occurrence of this phenomenon, likely in utero." The "this phenomenon" in this sentence is likely referring to WGD, but I recommend Authors to spell it out as WGD, even if the word WGD is repeated three times in a single sentence.
- 9) "Most tumours with WGD (42/56; 75%) had synchronous chromosomal gains (Fig. 4b), broadly in line with the distribution of gain patterns reported by the Pan-Cancer Analysis of Whole Genome (PCAWG) in tumours with WGD35." Authors are skipping Figure 4a

and directly jump to Figure 4a here. This style of presentation is confusing although I am not sure whether this journal requires Authors to refer to figures in an orderly manner.

- 10) “Additionally, a subset of enriched CNAs were specific to young-onset older-onset seminomas, such as chr8:45-129Mb and chr7:60-159Mb (Fig. 4c).” A typo: “specific to young-onset older-onset seminomas” -> “specific to young-onset seminomas.”

Shown below are scientific comments:

- 1) Supplementary Figure 1 shows bimodal age distribution of the age of diagnosis, and the text states, “with most seminomas being diagnosed between age 20y and 40y.” Authors are requested to add additional, color-coded curves for seminomas and NSGCTs to this figure so that READERS can verify the basis of your claim.
- 2) Whereas Authors discuss driver mutations of TGCTs, there is no mention to gain-of-function or loss-of-function mutations. Please refer to the gain/loss of functionality for key mutations as adequate.
- 3) Authors claim, “As expected, KIT mutations were seen only in seminomas and were mutually exclusive with KRAS/NRAS mutations (Supplementary Fig. 3). However, the mutual exclusivity of KIT against KRAS/NRAS shown in Supplementary Fig. 3 is unconvincing. In this figure, the NRAS(vertical)-vs-KIT(horizontal) cross point and the KRAS(vertical)-vs-KIT(horizontal) cross point both show only modest P-values that are not particularly stronger than most of other comparisons. Authors are requested to explain.
- 4) Authors stated, “Hypomethylated EP300 binding sites have been described in pediatric YSTs, which, together with our data, suggests alternate pathways through which EP300 can influence TGCT subtype (Ref# 13).” It is unclear as to why a feature of EP300 epigenetics reported for pediatric (Type I) YST is relevant to the current study, which focuses on adult-onset (Type II) GCTs. Authors are requested to explain.
- 5) Authors claim, “Overall, these data support the notion that functional integrity of non-coding elements is likely to play a role in TGCT oncogenesis.” However, the actual data support the absence of non-coding drivers. Only a single CTCF binding site was picked up under negative selection whereas none was found under positive selection. A relaxed cutoff value picked up two additional CTCF binding sites and the distal promoter of DLL1. Authors are requested to explain why they think that their analysis SUPPORTS the notion that non-coding elements play roles in TGCT oncogenesis. By the way, the CTCF site that was picked up under the strict criteria locates very close to the telomeric region. Authors are encouraged to examine possible risks of reporting functionality of near-telomeric CTCF sites and briefly comment about it, whether they found the location of this particular CTCF site still adequate or turned out to be risky.

- 6) Linking to Supplementary Figure 7 and Supplementary Table 11&12, Authors state, “In addition to established copy number alterations (CNAs), including amp4q12 (19% cases), amp1p36.32 (17% cases), and del9p24.3 (37% cases), spanning KIT, MDS2, and DMRT1, we identified 26 additional novel recurrent CNAs ... FLI1, MAF, and SMAD4 (Supplementary Fig. 2)” Authors are requested to explain the following:
- a. How many CNAs were established by your analysis? Are Supplementary Tables 11&12 present exhaustive lists?
 - b. Supplementary Figure 7 does not show amp1p36.32 (linked to MDS2). The link of del9p24.3 with DMRT1 is not shown, either. Amplification of 7q11.23 is linked to CDK6 in the text, but Supplementary Figure 7 also links this amplification to HIP1. Neither TP53AIP1 nor SMARCA2 is shown in Supplementary Figure 7 although they are mentioned in the text.
 - c. “... KIT amplifications were observed in NSGCT.” Where is this shown?
 - d. Authors referred to gene pairs PIK3CA-MCL1, RB1-FLI1, and MAF-SMAD4 as most significant driver interactions. However, Supplementary Figure 3 (which is wrongly referenced here as Supplementary Figure 2, as I pointed earlier) shows strong interactions between RB1-MEN1 and MEN1-FLI1. Please explain why these particular two pairs are omitted in the text.
 - e. Please mention to chromosomal locations of the key drive genes in the context of interaction. If all driver genes locate significantly separately, please state so. If there are pairs locating in close proximity to each other, it should be mentioned.
- 7) Authors describe about the GEL-TGCT-0056 case as an STM. I am not sure whether only one case of observation deserves focused discussion as presented in the current text.
- 8) “... with a general trend towards a lower proportion of subclonal mutations attributed to SBS5 and an increase of SBS18 ($p = 2.2 \times 10^{-16}$ and $p = 2.1 \times 10^{-11}$, respectively; test for trend in proportions).” Supplementary Figure 9 shows only two datum points for SBS18. Authors are requested to explain if the above claim is made based on the data shown in Supplementary Figure 9 (i.e., N=2) or other data are involved. If N=2, please justify the statistical procedure to obtain a meaningful p-value.
- 9) “... Analysis of the indel mutational spectra revealed a predominance of ID1 and ID2, both due to slippage during DNA replication²⁸.” Supplementary Figure 9 shows the presence of outliers in the data for ID1 and ID2. Please discuss how such outliers may or may not affect the conclusions.
- 10) “ ... ID3, a signature thus far linked with tobacco-smoke mutagens²⁸, was detected in all tumours.” Please clarify whether ID3 signature was detected in all TYPES of tumors, or

every single CASE of tumors involved in this study. ID3 shows large differences between tumor types, and Authors are requested to discuss this – namely, ID3 is typically detected only in tumors containing seminoma components. If Authors think otherwise, please explain.

- 11) “Deletion patterns characterised by ID6 and ID8 and arising from distinct mechanisms of DNA double-strand-break repair²⁸, were mutually exclusive.” This claim does not seem sufficiently supported by the data shown in Figure 3a. Please explain.
- 12) Supplementary Figures 11 and 12 need more explanation in the legends.
- 13) “Recent single cell analyses suggest that neonates possess a small pool of gonadal cells with characteristics of primordial germ cells (PGCs) (Ref# 33).” I suggest Authors to add “in their testes” at the end.
- 14) “Across all subtypes, mutations in driver genes including KIT, KRAS, and NRAS were relatively late events, occurring post-WGD and after corresponding copy number gains or other CNAs (Fig. 4a).” Figure 4a does not present KIT or NRAS. Please explain how Authors can draw the above conclusion from Figure 4.

In Figure 4a, the chromosome 4p region shows a distinctive peak. Does this region contain KIT? Similarly, in chromosome 6p, there is a region with reduced LOH compared to other part in the same chromosome, and it seems spanning over BAK1. Authors may want to comment on these.

- 15) “Most of the earliest events following WGD were gains and showed balanced representation across TGCTs (Fig. 4b), while later enriched events were specific to TGCT types, such as the 12q11 gain spanning and KIF21A restricted to NSGCTs.” Figure 4b does not present KIF21A although some of the 12q regions shown in this figure may include it. Please clarify.

Response to Reviewers:

We greatly appreciate the comments made by the Reviewers. We have responded to each of these in turn below and incorporated as many of the suggested changes as possible. We have significantly revised both the main text and supplementary materials, which we believe has substantially enhanced the manuscript's quality. Additionally, we have updated our results by including extended analysis on HLA-mediated immune disruption in TGCT and the temporal ordering of mutational events in testicular cancer genomes. Each comment is addressed below in red. Changes made to the manuscript have also been highlighted in red.

Reviewer #1 (Remarks to the Author)

This manuscript reports outcomes of a comprehensive WGS study on adult TGCT cases administered in seven centers located within the UK. Although the scientific contents described in this manuscript may be worth attention from investigators in the relevant fields of research, multiple issues prevent me from recommending the current manuscript for publication. I must start with an unusual comment – namely, the manuscript is organized very poorly. There are errors in the text that refer to wrong figures or supplementary figures. There are figures or supplementary figures that are not mentioned in the text even once. In one figure, critical symbols are flipped. These errors are too significant that affect my assigned task to examine the scientific merit thoroughly and fairly. Shown below is a list of such errors that I caught, but I am not sure whether my list is exhaustive, or some errors may have escaped my eyes.

We thank the Reviewer for their very thorough critique of the manuscript and for their helpful suggestions. We have taken the opportunity at the revision stage to go through the revised manuscript comprehensively, including correction of the errors listed above.

Response:

1) Authors state, "... however only SBS5 and the number of C>T mutations at NpCpG trinucleotides correlated with age ($p = 4.3 \times 10^{-8}$ and $p = 0.02$, respectively; Fig. 3c)." There is no panel C in Figure 3.

We thank the Reviewer for spotting this error. This has been corrected so that we refer to the correct Figure (which has now been moved to Supplementary - Figure S11). Please see Page 9, from line 256.

2) Supplementary Figure 2 is never CORRECTLY mentioned in the text. There is a WRONG mention to Supplementary Figure 2 in the context, "Mutual exclusivity analysis revealed ... FLI1, MAF, and SMAD4 (Supplementary Fig. 2)," but this should refer to Supplementary Figure 3.

This has been corrected so that we refer to Supplementary Figure 3. Please see Page 7, line 182.

3) “This approach revealed two structural variant (SV) signatures (S1, S2; Supplementary Figs. 14-15), present in both seminomas and NSGCTs.” This appears to be incorrect and should be Figs. 13-14.

This call out has now been corrected. Please see Page 10, from line 300.

4) “In all but one case (56/57), tumours were shown to have undergone WGD (Supplementary Fig. 15).” Supplementary Figure 15 shows only one tumor with WGD (red circle) whereas all other datum points shown with blue circle symbol are Diploid

We thank the Reviewer for spotting this error. This Figure has been updated.

5) “This is in stark contrast to most solid cancers, where WGD events are broadly distributed throughout clonal evolution and likely stochastic (Fig. 5a).” Authors skipped any mention to Figure 4 before referring to Figure 5 here. The major problem here is that Figures 5b, 5c, or 5d are never mentioned in the text.

We have now re-ordered the Figures and called them out in the correct order. Please see Page 11, line 320 with regards to the correction of this specific error.

6) “A probabilistic timing model was used to reconstruct the order of acquisition of recurrent genomic aberrations, including WGD, enriched CNAs, and putative driver mutations across all TGCTs and within each of seminomas and NSGCTs (Methods, Fig. 3b,c).” This sentence should refer to Figures 4b and 4c.

We thank the Reviewer for this comment. The call out to the Figures has accordingly been corrected. Please see Page 12, line 353.

7) Supplementary Figure 17 is never mentioned in the text.

We thank the Reviewer for raising this. Whilst Supplementary Figure 17 is not mentioned in the main text, we do refer to this Figure in the Methods. The retention of this Figure is important as it supports the use of a specific threshold in our analysis.

The following issues may not be errors, but Authors are requested to explain and/or make adequate corrections:

1) The text describes that 60 tumor samples were analyzed, but Figure 1b presents only 58 tumors. The figure correctly present 39 primary seminomas, but it shows only 15 primary NSGCTs although it should be 16. The figure shows only 4 metastatic tumors, but the text refers to 5 metastatic tumors.

Whilst 60 tumour samples were analysed, in Figure 1b two samples not generated via PCR-free library preparation were excluded. These samples were included in the Figure as we observed elevated numbers of single-base substitutions and indels. One of the excluded samples was a metastatic NSGCT and the other a NSGCT primary. Despite the elevated counts we could still robustly determine mutational signatures and detect large-scale mutation events (copy number, structural variants, etc.).

We agree with the Reviewer that the Figure legend needs more detail and have now clarified this.

2) Author states, “Eight genes were significantly somatically mutated,” but these genes are not listed. I had to take a look at the supplementary tables to figure out these eight genes. Please list these genes in the text.

These genes are now listed in the main text. Please see Page 5, line 133.

3) Authors state, “Eight genes were significantly somatically mutated, including KIT (24%) and its downstream mediators KRAS (7%) and NRAS (4%) (Fig. 1b). But Figure 1b only shows KIT mutations. Authors are requested to add KRAS and NRAS mutation profiles on Figure 1b.

We thank the Reviewer for this suggestion. We have updated Figure 1b to include information about KRAS/NRAS mutations.

4) Authors may want to consider substituting the subtitle “Focal amplifications” with “Extrachromosomal DNA.”

The subtitle has been changed. Please see Page 8, line 230.

5) “Though not observed in NSGCTs within the GEL cohort with a predominant YST component, both cases with SBS18 harboured a YST component.” Please specify which cases mentioned here as “both cases.” The preceding sentences refers to one case, but it is not easy to identify the other case mentioned here.

We thank the Reviewer for this point and have now clarified the text so that our description is clearer - SBS18 was detected in two tumours and in one of these cases, SBS18 was the predominant signature. Please see Page 9, from line 259.

6) Supplementary Figure 9 shows two pictures in each panel. I am assuming that the right panel is for non-clonal mutations and the left is clonal mutations. The legend should clarify this matter.

The left hand panel showed signatures identified in the 57 independent TGCT samples; the right hand panel showed signatures identified in 60 genomes, including multiple samples from a single patient. The purpose was to indicate the robustness of signature detection. However, we

agree with the Reviewer that showing both sets of results is confusing for the reader and have amended Supplementary Figure 9 and the accompanying legend so that only one set of analyses is shown giving an overview of signatures across all 60 genomes. We have also now included Supplementary Tables summarising the mutational signature repertoires.

7) "... and associated with TP53 drivers (RefSig.R2) and unclustered deletions up to 100kb associated with BRCA2 mutations (RefSig.R5), respectively (Ref# 32)." The titles shown in Supplementary Figure 14 are truncated (right columns) and cannot be verified for their links to TP53 or BRCA2.

The Figure has been amended. We have also updated the text to make it clear that these associations were previously described in another study. Please see Page 10, from line 306.

8) "We observed a median of just ~9 substitutions (range 0-375) occurring prior to WGD, and in seven cases we did not observe any pre-WGD substitutions, supporting early occurrence of this phenomenon, likely *in utero*." The "this phenomenon" in this sentence is likely referring to WGD, but I recommend Authors to spell it out as WGD, even if the word WGD is repeated three times in a single sentence.

We thank the Reviewer for this comment and have amended the sentence to make this point clearer. Please see Page 11, line 320.

9) "Most tumours with WGD (42/56; 75%) had synchronous chromosomal gains (Fig. 4b), broadly in line with the distribution of gain patterns reported by the Pan-Cancer Analysis of Whole Genome (PCAWG) in tumours with WGD35." Authors are skipping Figure 4a and directly jump to Figure 4a here. This style of presentation is confusing although I am not sure whether this journal requires Authors to refer to figures in an orderly manner.

We agree with the Reviewer that this is potentially confusing and we have reordered the Figures and corrected all Figure callouts accordingly throughout the manuscript.

10) "Additionally, a subset of enriched CNAs were specific to young-onset older-onset seminomas, such as chr8:45-129Mb and chr7:60-159Mb (Fig. 4c)." A typo: "specific to young-onset older-onset seminomas" -> "specific to young-onset seminomas."

We thank the Reviewer for spotting this typographical error and have corrected it. Please see Page 13, line 380.

Shown below are scientific comments:

1) Supplementary Figure 1 shows bimodal age distribution of the age of diagnosis, and the text states, "with most seminomas being diagnosed between age 20y and 40y." Authors are requested to additional, color-coded curves for seminomas and NSGCTs to this figure so that READERS can verify the basis of your claim.

Previously Supplementary Figure 1 showed the age distribution for patients with seminomas only. We have now updated the Figure showing two age distributions, for patients with both seminomas and NSGCTs. These updated data broadly align with observed trends in seminoma diagnosis (see PMID: 31871817 for example), wherein a significant peak in seminoma diagnosis occurs among men in their second and third decades of life. Relatively fewer occurrences of seminomas are noted in older men, which is consistent with our findings.

2) Whereas Authors discuss driver mutations of TGCTs, there is no mention to gain-of-function or loss-of-function mutations. Please refer to the gain/loss of functionality for key mutations as adequate.

We have updated the text throughout to make explicit the roles of the various putative drivers identified. Please see Page 6, lines 142-143, 152 etc.

3) Authors claim, “As expected, KIT mutations were seen only in seminomas and were mutually exclusive with KRAS/NRAS mutations (Supplementary Fig. 3). However, the mutual exclusivity of KIT against KRAS/NRAS shown in Supplementary Fig. 3 is unconvincing. In this figure, the NRAS(vertical)-vs-KIT(horizontal) cross point and the KRAS(vertical)-vs-KIT(horizontal) cross point both show only modest P-values that are not particularly stronger than most of other comparisons. Authors are requested to Explain.

We thank the Reviewer for raising this point. We have updated the text so that *only* significant co-occurring or mutually exclusive mutations are highlighted. Please see Page 7, from line 181.

4) Authors stated, “Hypomethylated EP300 binding sites have been described in pediatric YSTs, which, together with our data, suggests alternate pathways through which EP300 can influence TGCT subtype (Ref# 13).” It is unclear as to why a feature of EP300 epigenetics reported for pediatric (Type I) YST is relevant to the current study, which focuses on adult-onset (Type II) GCTs. Authors are requested to explain.

We agree with the Reviewer that this comment is not relevant to the *EP300* driver finding and have removed it from the text.

5) Authors claim, “Overall, these data support the notion that functional integrity of non-coding elements is likely to play a role in TGCT oncogenesis.” However, the actual support the absence of non-coding drivers. Only a single CTCF binding site was picked up under negative selection whereas none was found under positive selection. Relaxed cutoff value picked up two additional CTCF binding sites and the distal promoter of DLL1. Authors are requested to explain why they think that their analysis SUPPORT the notion that non-coding elements play roles in TGCT oncogenesis. By the way, the CTCF site that was picked up under the strict criteria locates very close to the telomeric region. Authors are encouraged to examine possible risks of reporting functionality of near-telomeric CTCF sites and briefly comment about it, whether they found the location of this particular CTCF site still adequate or turned out to be risky.

We agree with the Reviewer that it is perhaps too strong to say that there is evidence of a role for non-coding elements in TGCT oncogenesis based on the evidence provided here and have accordingly removed this statement.

TGCT has a very low mutation burden and the test in OncodriveFML for “negative selection” is those elements in which there are fewer functional mutations observed (based on CADD scores) than expected based on local trinucleotide context. Whilst one interpretation is that significant results indicate negative selection, another could be that the region happened to deviate from the background model for another reason (e.g. because the local mutational processes differed significantly from the rest of the genome). We have now made this caveat explicit in the description of our Methods and in the main text. Please see Page 8, from line 210.

6) Linking to Supplementary Figure 7 and Supplementary Table 11&12, Authors state, “In addition to established copy number alterations (CNAs), including amp4q12 (19% cases), amp1p36.32 (17% cases), and del9p24.3 (37% cases), spanning KIT, MDS2, and DMRT1, we identified 26 additional novel recurrent CNAs ... FLI1, MAF, and SMAD4 (Supplementary Fig. 2)” Authors are requested to explain the following:

a. How many CNAs were established by your analysis? Are Supplementary Tables 11&12 present exhaustive lists?

Supplementary Tables 11 & 12 do not present exhaustive lists. Tables 10 (previously table 11) summarise all recurrent arm-level and focal amplifications and deletions detected by GISTIC2 from across all CNA events identified in the GEL TGCT cohort. Table 11 (previously) provides a summary of annotated focal segments. In total, 3,085 individual CNA events were identified across the 57 individual samples using the Battenberg copy number caller (including only the highest cellularity multi-region sample for this count). While the restrictions of working with NHS data mean that only summary level analyses can be shared outside of the secure Genomics England (GEL) Research Environment (RE), individual-level CNAs can be found within the GEL RE in /published_data_archive/paper_reviews/paper_review_RR67 or

/published_data_archive/paper_data/paper_data_RR67.

b. Supplementary Figure 7 does not show amp1p36.32 (linked to MDS2). The link of del9p24.3 with DMRT1 is not shown, either. Amplification of 7q11.23 is linked to CDK6 in the text, but Supplementary Figure 7 also links this amplification to HIP1. Neither TP53AIP1 nor SMARCA2 is shown in Supplementary Figure 7 although they are mentioned in the text.

We have now corrected this omission.

c. “... KIT amplifications were observed in NSGCT.” Where is this shown?

This is shown in Figure 1b (see KIT row where red boxes correspond to KIT amplifications). We have edited the text to clearly indicate the relevant supporting Figure and data to the reader. Please see Page 6, from line 172.

d. Authors referred to gene pairs PIK3CA-MCL1, RB1-FLI1, and MAF-SMAD4 as most significant driver interactions. However, Supplementary Figure 3 (which is wrongly referenced here as Supplementary Figure 2, as I pointed earlier) shows strong interactions between RB1-MEN1 and MEN1-FLI1. Please explain why these particular two pairs are omitted in the text.

We have corrected the Figure callout and updated the text to refer to all of the most significant co-occurring and mutually exclusive drivers. Please see Page 7, from line 181.

e. Please mention to chromosomal locations of the key drive genes in the context of interaction. If all driver genes locate significantly separately, please state so. If there are pairs locating in close proximity to each other, it should be mentioned.

We have now updated the text to indicate which pairs are inter- versus intra-chromosomal. Please see page 7, line 186.

7) Authors describe about the GEL-TGCT-0056 case as an STM. I am not sure whether only one case of observation deserves focused discussion as presented in the current text.

Whilst this is just one observation, we believe it is worth commenting on the features of this case which are found to be unique within the cohort. TGCT with somatic malignant transformation is a rare phenomenon within a relatively rare cancer and in the metastatic setting, as in this case, outcomes are often poor. As such, we strongly believe this single case merits discussion.

8) "... with a general trend towards a lower proportion of subclonal mutations attributed to SBS5 and an increase of SBS18 ($p = 2.2 \times 10^{-16}$ and $p = 2.1 \times 10^{-11}$, respectively; test for trend in proportions)." Supplementary Figure 9 shows only two datum points for SBS18. Authors are requested to explain if the above claim is made based on the data shown in Supplementary Figure 9 (i.e., $N=2$) or other data are involved. If $N=2$, please justify the statistical procedure to obtain a meaningful p-value.

On reflection, we agree with the Reviewer that there are insufficient data points to justify the statement about SBS18 included in the main text and have removed this.

9) "... Analysis of the indel mutational spectra revealed a predominance of ID1 and ID2, both due to slippage during DNA replication²⁸." Supplementary Figure 9 shows the presence of outliers in the data for ID1 and ID2. Please discuss how such outliers may or may not affect the conclusions.

Here, where we say that there is a predominance of ID1 and ID2, we are referring to the fact that they are the most prevalent signatures detected across the cohort. Even excluding outlying samples this remains the case. This is also consistent with previous reporting on indel signatures. Unlike the non-ubiquitous signatures, ID1 and ID2 are found in most cancer samples.

10) “ ... ID3, a signature thus far linked with tobacco-smoke mutagens²⁸, was detected in all tumours.” Please clarify whether ID3 signature was detected in all TYPES of tumors, or every single CASE of tumors involved in this study. ID3 shows large differences between tumor types, and Authors are requested to discuss this – namely, ID3 is typically detected only in tumors containing seminoma components. If Authors think otherwise, please explain.

We thank the reviewer for highlighting this ambiguity. The text should have read ‘*..all types of TGCT*. We have since updated the text (see section ‘*Complete repertoire of mutational signatures*’). We have also included supporting tables (S18-S22) showing the complete repertoire of SBS, ID, DBS, CNA and SV signatures.

11) “Deletion patterns characterised by ID6 and ID8 and arising from distinct mechanisms of DNA double-strand-break repair²⁸, were mutually exclusive.” This claim does not seem sufficiently supported by the data shown in Figure 3a. Please explain.

Please also see response to point 10 above - we have now included supporting Tables showing the complete repertoire of SBS, ID and DBS signatures. These data support this claim and are now called out within the text. Please see Page 10, from line 277.

12) Supplementary Figures 11 and 12 need more explanation in the legends.

We thank the Reviewer for this comment. The legends have accordingly been updated.

13) “Recent single cell analyses suggest that neonates possess a small pool of gonadal cells with characteristics of primordial germ cells (PGCs) (Ref# 33).” I suggest Authors to add “in their testes” at the end.

We thank the Reviewer for this suggestion and have updated the sentence. Please see Page 11, line 330.

14) a) “Across all subtypes, mutations in driver genes including KIT, KRAS, and NRAS were relatively late events, occurring post-WGD and after corresponding copy number gains or other CNAs (Fig. 4a).” Figure 4a does not present KIT or NRAS. Please explain how Authors can draw the above conclusion from Figure 4.

The Reviewer is correct and there is a mistake in the figure reference. The relative ordering inferred using our model and shown in the left hand panels of Fig. 4b and c show that these

driver mutations are relatively late events. Our reference to this figure has now been corrected. Please see Page 13, from line 388.

14) b) In Figure 4a, the chromosome 4p region shows a distinctive peak. Does this region contain *KIT*? Similarly, in chromosome 6p, there is a region with reduced LOH compared to other part in the same chromosome, and it seems spanning over *BAK1*. Authors may want to comment on these.

The Reviewer is correct and the peaks observed in Figure 4a spanning 4p and 6p contain *KIT* and *BAK1*, respectively. However, we have only highlighted enriched events as determined using our probabilistic model. This has now been clarified in the Figure legend.

15) “Most of the earliest events following WGD were gains and showed balanced representation across TGCTs (Fig. 4b), while later enriched events were specific to TGCT types, such as the 12q11 gain spanning and *KIF21A* restricted to NSGCTs.” Figure 4b does not present *KIF21A* although some of the 12q regions shown in this figure may include it. Please clarify.

Here the text should refer to Figures 5b and 5c where the relative timing of *KIT*, *KRAS* and *NRAS* driver mutations is shown. This has now been corrected. Please see page 13, from line 373.

Reviewer #2 (Remarks to the Author)

Lethlobhair et colleagues present a thorough analysis of whole genome sequencing data of from testicular germ cell tumors (TGCT). While not an exceedingly large study by today's standards (60 tumors), and smaller than the TCGA study on TGCT (~130), it is nevertheless the most comprehensive WGS analysis of TGCT thus far. While there are no real breakthrough insights into driver mechanisms or similar - despite a few proposed new driver genes/mutations - the analyses appear generally to be carried out in an expertly manner and the results should constitute a useful contribution to this field of research.

Specific comments:

1) Regarding newly identified drivers (KLF4, RAC1, EP300) - how come these were not found e.g. in the larger TCGA cohort? This is worth discussing. Are they mutated to some extent without reaching significance?

A nonsynonymous *EP300* mutation was identified in one TCGA case (and the Intogen driver pipeline requires candidate drivers to be mutated in at least two different tumours). The *EP300* mutation is also only found in a small number of GEL and MSK cases so it is likely to be a rare or low frequency TGCT driver. A similar observation accounts for the *KLF4* and *RAC1* findings. This is now discussed in the text, see page 15, from line 445.

2) Intuitively, I am somewhat sceptical that negative selection in ncDNA can be reliably detected in such a small low-burden cohort (as signals of negative selection have proved very limited even in coding DNA and even in large cohorts). I'm not saying these results are necessarily wrong, but selection is after all difficult to assess - so I think the authors should consider using a more cautious statement (now: "is likely to"). Also in discussion "evidence" for this is mentioned, which I personally think is a too-strong word: it is impossible to gain "evidence" of selection in these analyses: a low P value simply means there is a deviation the modelled expectation and the model is always wrong to some extent (the question is just how much).

TGCT has a very low mutation burden and the test in OncodriveFML for "negative selection" is those elements in which there are fewer functional mutations observed (based on CADD scores) than expected based on local trinucleotide context. While one interpretation is that significant results indicate negative selection, another could be that the region happened to deviate from the background model for another reason (e.g. because the local mutational processes differed significantly from the rest of the genome). We have now made this caveat explicit in the main text.

We agree with the reviewer that this finding should be presented more cautiously in the text and have now done so; see page 8 from line 210.

3) I believe WGD and diploid are incorrectly labelled in Fig. S15. It is not obvious how this classification was carried out and information in the Fig. S15 is limited, although there are a few notes in the Methods supplement about a regression line by Dentro et al. It's hard to judge how reliable this classification is. Minimum, perhaps some clarification can be added to the legend to provide some general guidance about how this was done.

We thank the Reviewer for spotting this error. This Figure has now been corrected.

4) I don't see how Fig. 4b supports the claim about synchronous chromosomal gains? Please clarify.

We apologise for the confusion. The Figure callout has been corrected and we now refer the reader to the Methods also where our approach for timing mutations has been described. Please see Page 12, from line 342.

5) In presenting the mutational timing results, which are rather complex, perhaps the limitations (and underlying principles) can be shortly mentioned in just a sentence or so - e.g. limitations regarding deletions and mutations in non-CNA or deleted regions.

We thank the Reviewer for this comment and have now added a brief statement preceding the presentation of the mutational timing results outlining the underlying principles of our analysis. Please see Page 11, from line 314.

6) The supplement is weirdly formatted with blank pages and legends often appearing on the next page - or being split between two pages. Also, some of the pictures are not readable due to pixelated text e.g Fig. S8.

We thank the Reviewer for this comment. Unfortunately, an issue affected the uploaded Figures during the original submission, after they were merged, outwith our control. We hope this has now been resolved and that there are no further issues with formatting or image resolution.

Reviewer #3 (Remarks to the Author)

This manuscript comes from Genome England and describes the whole genome landscape of TGCT. This manuscript was incredibly frustrating to read, as the Figures and Supplemental Figures are poorly labeled, they appear to have cut and pasted sections from other papers in the Methods section that do not apply here, and multiple other errors were made (i.e. Figure 5 comes before Figure 4 in text, and Figures 5b, 5c, and 5d are never referred to). To note, it is not be clear that the origin of TGCT is the primordial germ cell, as recent studies have suggested it could be the gonocytes, so the authors need to adjust their language about the cells of origin and not be quite so specific – e.g. the term fetal germ cells, which covers both possibilities.

In general, the manuscript presents interesting whole genome data that explores the mutational signatures and timing of events during tumor development. However, much of what is shown is not entirely novel, such as the timing of KIT mutations after WGD, but the figure showing the timing is elegant and of interest. They also present a deep dive into mutational signatures and demonstrate HLA loss, which provide novel data in the field. However, they have several essentially anecdotal points in the results, which could easily be removed, so that they can focus on novel elements of their data. They also make a number of off hand comments that are not clearly supported by the data.

We thank the Reviewer for their positive comments. We have now made substantial changes to the manuscript in line with comments from the Reviewers, placing a greater emphasis on the more novel aspects of the study and in particular extending our analyses of HLA loss and timing.

We have amended the part of the text where we make reference to a specific cell-of-origin. Please see Page 4, line 88-89.

1. It is not quite correct to say a comprehensive description of the genomic landscape of TGCT is lacking. Beyond the TCGA on TGCT (PMID: 29898407), an overlapping author group published an in-depth examination of TGCT in 2022 (PMID: 35953478). The latter paper also included whole genome sequencing data, although the focus of the analysis was somewhat different.

We have now amended the text to say that a '*... a comprehensive description of the whole genomic landscape of TGCT is largely missing from the literature*'. Please see Page 4, line 95.

The TCGA study, while including more samples, comprised only exome sequencing. The more recent study on TGCT published in 2022, while presenting whole genome data, only described eight individual adult patients with TGCTs and one of these 8 cases was derived from a precursor GCNIS (germ cell neoplasia in situ) lesion. We now specifically acknowledge this.

2. The bimodal peak with the secondary peak occurring in the late 50s, does not seem consistent with what one would consider classic TGCT. Is this epidemiology reflective of the distribution of TGCT diagnosis in the UK?

The data previously presented in Supplementary Figure 1 only showed the age distribution for participants with testicular seminomas. We have since updated this Figure to show the distribution for NSGCTs also. Supplementary Figure 1 is created by modelling age distribution in the cohort using the R package multimode.

With regard to the age distribution within the UK age – specific incidence rates rise steeply from around the age of 10-14, peak at 30 – 34 and then rapidly decline (CRUK cancer stats) - this is very much what the distribution of our NSGCT cases showed. Seminoma is known to have a bimodal distribution as our data has shown. But we also accept that overall our numbers are small.

3. Table S3 is labelled as ‘Table S3. List of all substitutions and indels called across the GEL TGCT cohort.’ To note, please fix the typo. The Table is 1239 lines long and 1234 of the variants occur just once, so it can’t include all substitutions and indels, as described in the text.

We have re-checked Table S3 and it is correct. In total, 1,239 sites are listed; 1,238 of these lines correspond to unique sites. One site (12:25245350) is listed twice where two different ALT alleles were identified in the data.

We have also corrected the table legend as this should read instead ‘**Table S3. List of all nonsynonymous substitutions and indels called across the GEL TGCT cohort and analysed as part of the intOGen pipeline**’. While the restrictions of working with NHS data mean that only summary level analyses can be shared outside of the secure Genomics England (GEL) Research Environment (RE), the full list of individual-level SNVs and indels can be found within the GEL RE in /published_data_archive/paper_reviews/paper_review_RR67 or /published_data_archive/paper_data/paper_data_RR67.

4. Should Table S5 precede Table S4, as it is the data from the GEL cohort, and Table S5 contains the combined cohorts. What the column CGC_Cancer_Gene, which is entirely FALSE in both tables indicates is not clear, since many of the genes are cancer genes.

The Supplementary Tables have now been reordered correctly.

“CGC_Gene” refers to whether the gene is in the cancer gene census, and “CGC_Cancer_Gene” is if the given cancer type (e.g. “TGCT”) is one of the somatic cancer types for that gene listed in the cosmic cancer gene census. We agree that this is confusing and have now included a key in Table S5 explaining these column names.

5. To note that their analysis did not show significance in mutual exclusivity between *KIT* and *KRAS/NRAS*, so although true, not sure of the meaning of their analysis in Supp Fig 3. Additionally, they mention that *KIT* mutations are clustered in intracranial TGCT, but not in classic TGCT?

We have now removed mention of mutual exclusivity between *KIT* and *KRAS/NRAS* as this finding was not statistically significant.

We have updated the text to make it clear that *KIT* mutations also cluster on exon 17 in testicular seminomas. Please see Page 5, from line 137.

6. Figure 2 – Given that the Figure 2 includes the commonly amplified and deleted genes at the top, which are not referred to until later in the manuscript, it is confusing for the reader. Perhaps the sections of text can be reorganized so that the reader isn't trying to parse through trying to follow the Figure well before that section of text.

We agree with the Reviewer that this is confusing and have significantly re-organised the text to improve the readability of the manuscript.

The commonly amplified and deleted genes listed in the figure are not discussed in the text – e.g. *BCOR*, *SMAD4*, *MEN1* and *CDH17*. Additional information needs to be provided about the genes that they are highlighting in Figure 2. Presumably they just select genes from their GISTIC results to include in Figure 2 based on their description in 8.3.3 (Methods), but it is not stated. Usually the copy number analysis prior to running GISTIC is used to determine most commonly deleted/amplified genes (in this case Battenberg per-tumor segmentation output, which needs to be referenced). Were cut-offs for deletion and amplification used for this analysis.

We have updated the GISTIC figure so that only genes discussed in the text are listed in the figure; otherwise a more complete annotation is available from the Supplementary Tables 10 & 11.

We have now made it clear in the text the copy number analysis was performed using Battenberg prior to running GISTIC (see page 6, line 164).

The cutoffs used when running GISTIC are described in detail in the Methods section of the manuscript which we have also referenced.

Their criteria for calling a missense significant is not clear – e.g. what criteria did the two missense variants in *EP300* meet (a gene in which you expect loss of function mutations) to be included as candidate driver mutations? What is the double color (for *KIT* mutations mainly, missense/amplified) – needs to be clear. What does multi-hit mean?

Driver gene identification was performed using the IntOGen pipeline, which incorporates seven complementary algorithms. The specifics of this pipeline can be found in the Methods. The *EP300* mutation is found in a small number of GEL and MSK cases and a nonsynonymous *EP300* mutation was identified in one TGCA case. The Intogen driver pipeline requires candidate drivers to be mutated in at least two different tumours). This is likely to be a rare or low frequency TGCT driver. A similar observation accounts for the *KLF4* and *RAC1* findings. This is now discussed in the text, see page 15, from line 445.

We have updated the legend to explain the meaning of the multi-hit and the double colour blocks.

7. Table S8 – Their analysis suggests that 27% and 18% of non-seminomas in the TGCT TGCA analysis have *KIT* and *KRAS* mutations, which is not the case (see Figure 2 from the manuscript and below). It's pretty clear that *KIT/KRAS* mutations are associated with seminomas. Maybe they have entirely mislabeled their Table – should it be seminomas? They also identify *KIT* mutations in several of the NSGCT in their own cohort (see Figure 2), again as this phenomenon is not reported in other cohorts, they need to clarify – are they mixed/combined tumors.

We thank the Reviewer for spotting this typographical error in the Supplementary Table label; this should read 'seminomas' and has now been corrected.

We have added to the text to clarify our finding of *KIT* amplifications in NSGCTs. Please see Page 6, from line 173.

8. Hard to know what to make of their comment about *FAT4*, since it does not show up at all in their cohort.

It is likely that *FAT4* is a rare or low frequency driver and that is why it is not seen in the GEL cohort. We have now added a statement to this effect see page 15; line 445. Worth noting, although *FAT4* drivers weren't identified in the MSK-MET dataset, we did detect three nonsynonymous *FAT1* mutations. Moreover, a recent study (PMID: 37149691), now referenced in the manuscript, has reported mutations in the *FAT* gene family, including *FAT4*, in mixed malignant GCTs occurring during childhood and adolescence.

10. Page 8, middle - It is not at all clear how the authors can state that 'functional integrity of non-coding elements is likely to play a role in TGCT oncogenesis'. Based on what data? They show one significant CTCF site, and a few others that they have picked up – is that more than they would see based on the number of analyses that they have done by chance alone? Certainly it's not enough to support the statement they make.

We have removed this statement but still present the data supporting a putative role for mutations in non-coding regions in TGCT.

TGCT has a very low mutation burden and the test in OncodriveFML for “negative selection” is those elements in which there are fewer functional mutations observed (based on CADD scores) than expected based on local trinucleotide context. While one interpretation is that significant results indicate negative selection, another could be that the region happened to deviate from the background model for another reason (e.g. because the local mutational processes differed significantly from the rest of the genome). We have now made this caveat explicit in the main text.

11. Why 12p not show up at all in their GISTIC analyses? For the most common amplification in TGCT not be identified in their GISTIC analysis seems quite odd. Similarly, why did i12p not show up as part of the focal amplification analysis? Section 8.5 of the methods suggests that they did look for it.

A recurrent arm-level gain of 12p was found by GISTIC and is reported in Supplementary Table 10 GISTIC2 recurrent CNA segment summary. The legend for Supplementary Figure 7 did not make it clear that only focal events were shown. We have updated the legend of the GISTIC figure (now Supplementary Figure 3) to clarify this.

12. Most of the Supplemental Tables have undefined columns. For example, what do d.seg, rate.factor, d.bg, d.obs.exp and RD mean in Table S13. A key to the different column name should be provided throughout the Tables.

We have updated the Supplementary Tables so that undefined columns that are not easily interpretable are now clearly defined.

13. The discussion about the single tumor that has a somatic-type malignancy (Pages 9-10) does not add much to the manuscript and can be easily removed.

We respectfully request to keep this brief discussion in the main manuscript. Although this is a single case, we believe that the scientific value merits retaining this text. As few as 2.5% of testicular germ cell tumours undergo somatic malignant transformation and having material from a metastatic site for WGS represents a unique opportunity. See also response to Reviewer #1, Point 7 above.

14. In the mutational signature, there are several sets of data quoted not shown, including the differences by KIT mutation status, GEL-TGCT-0038, all of the other tumors about which there is specific discussion and supporting statement such as ID6 and ID8 are mutually exclusive. The point of this section is not clear, as it is just anecdotally discussing specific tumors (without showing their signatures). It is important that they show the SBS signature breakdown by tumor, especially as they are discussing various signatures as potentially of interest – e.g. SBS32. As that signature is seen in a very small proportion overall, whether or not it is relevant is unclear.

We agree with the Reviewer and have now included the complete repertoire of SBS, ID, DBS, CNA and SV signatures extracted across all GCT samples as Supplementary Tables 18-22.

What are they trying to show in Supplementary Figure 11, and how does it relate to the main text? It's not mentioned anywhere. A large component of Supplementary Figure 11 is not labeled (below the Figure Legend). What do the headers on the figures mean? It would be helpful if they provide more explanation (as in the Nature source paper).

On reflection, we agree that this Figure is confusing and this is an issue that other Reviewers have raised. We now present supporting data for the CN signatures in the form of a Table as opposed to a Figure (see Supplementary Table 21).

15. It is very difficult to follow the discussion about the copy number signatures as the supporting data are not presented (or maybe it's in the uninterpretable Supplemental Figure 11).

We have now included supporting data (see point 14 above).

16. The concluding sentence on Page 12 does not entirely make sense. Two signatures are referenced that are associated with TP53 mutations and BRCA2 mutations, neither of which are found in TGCT. Lots of detail are provide in the methods about extracting structural variants, although it is not clear how much was actually included in the manuscript – e.g. sections 9.4 – SV hotspots – 9.4.1 - Relationship between genomic features and SV rates – 9.4.2 - Permuting SVs - 9.4.3 Identifying SV hotspots – where are these data presented?

Regarding the first point, we have updated the text to make it clear that TP53 and BRCA2 represent previously reported associations (see page 10, from line 305).

The Methods sections highlighted by the Reviewer all describe different steps in the approach used to identify structural variant hotspots which is a permutation-based approach that considers genomic features associated with structural variation occurrence. We have now made it clear that these sections describe different steps within a single approach. The discovery of novel SV hotspots is reported in the main text (see section '*Hotspots of structural variation*').

17. Supplemental Figure 15 – is the labeling of WGS vs Diploid reversed, as the Figure suggests only one sample has WGD.

We thank the Reviewer for spotting this error. This figure has now been corrected.

18. The text skips from Figure 3 (page 11) to Figure 5 (page 13) – did the authors reverse Figure 4 (Page 15) and 5 in the submission? Why does it say Figure 4b at the end of page 13 – they mean Figure 5b.... No reference to Fig. 5c, 5d are in the text. What are they trying to show with these Figures?

We thank the Reviewer for spotting these inconsistencies. The figures have been re-organised and are now called out correctly.

19. The section mid-13 again discussing specific cases in a relatively anecdotal fashion, does not seem to contribute much to the manuscript.

While on first reading this may seem anecdotal, these details are quite pertinent. A recent publication (PMID: 35953478) reported that genome duplication universally arose very early across TGCT. Here we present the first evidence for a disease subgroup where this is not the case. All cases within this subgroup have a clinical history of bilateral TGCT or retractile/undescended testes and we believe this finding merits reporting and further investigation in a larger and/or targeted patient cohort.

We have amended the text to better communicate the value of this finding. Please see Page 11, from line 322

20. Why is the isochromosome section different than the focal amplification section?

When considering isochromosome formation, we were not just looking at 12p gains or at amplifications but at scenarios where there was evidence of at least two excess copies of 12p relative to 12q gained through the centromere (canonical formation) or otherwise (non-canonical). We have referenced the Methods within the main text to make this point apparent.

Why do they not include any supporting data? Supplementary Table 12 refers to the GISTIC data, which as noted above does not show any i12p?

While the restrictions of working with NHS data mean that only summary level analyses can be shared outside of the secure Genomics England (GEL) Research Environment (RE), individual-level CNAs supporting the isochromosome analysis specifically can be found within the GEL RE in /published_data_archive/paper_reviews/paper_review_RR67 or /published_data_archive/paper_data/paper_data_RR67.

The authors state that their finding contradicts prior studies that used i12p diagnostically for TGCT - the reference they use is from 1989. It is not used currently in that way. Do they find any target genes when they are evaluating the i12p? What genes are amplified is of interest, given the on-going discussion about the target genes of i12p.

We agree with the Reviewer that the diagnostic i12p reference is dated and have removed this. Please see Page 7, from line 189. We did not find any additional genes of interest beyond those already mentioned within the manuscript and already established in the literature.

21. In Supplemental Figure 16, what do the abbreviations mean? They need to be detailed in the Figure legend.

The Figure legend has now been updated to include more detail.

22. How did the authors come up with 17 candidate driver genes? Table 2 would suggest 16, if you include the genes without any mutations in this sample set (which seems improbable). What evidence is there that PTMA could be a target for therapeutic intervention? (Just seems a throwaway sentence) It's an immune function gene.

Supplementary Table 6 lists the driver mutations identified using a well established approach (IntOGen driver discovery); 17 candidate drivers were identified in total across the GEL and TCGA cohorts. We agree with the Reviewer re *PTMA*, and have removed the comment regarding the *PTMA* gene from the manuscript.

23. Finding SBS18 in one sample is not clearly indicative of an important event.

We have amended the text as our previous description of signatures was unclear. SBS18 was detected in two tumours with similar histologies and in one of these cases SBS18 was the predominant signature.

Minor Comments

1. A few grammatical notes – ‘While’ is a measure of time, and should be used as such. ‘Although’, ‘however’ and ‘whereas’ can be used instead. ‘These’, ‘this’ or ‘there’ should always be defined by a noun.

We thank the Reviewer for these notes and have now corrected the text accordingly.

2. Page 6, end – Sentence starting ‘However,.....’ doesn’t follow as however, doesn’t make sense.

We have now rewritten this section of the paper. Please see section ‘*Identifying subtype specific driver mutations*’

3. Supplementary Figure 8 is not readable.

We have updated all Supplementary Figures to improve resolution.

4. Top of Page 8 – Sentence starting ‘We assessed.....’. It is not clear that the authors are generally referred their findings (17% of what denominator) and not PTMA specifically, so the sentence needs to be clarified.

We have updated the text so that the denominator is now clear. Please see Page 6, from line 158.

5. Supplemental Figure 9. What is the difference between the left and right columns? Some of the signatures are repeated so it's not clear at all. Additionally, what do the numbers below the gray/white blocks mean? (not explained in figure legend)

Our cohort of 60 samples included 4 multi-region samples sequenced from the same individual. The four multi-region biopsies showed a high concordance of signatures across the 57 individuals (left hand column in previous Supp Fig 9) and the 60 samples including multi-regions from one individual (right hand column in previous Supp Fig 9). We agree that this is confusing.

6. Figure 3c is referred to in the text, and suggests a correlation with age, which is not shown in Figure 3b (no Figure 3c).

We have corrected the Figure so that these correlations are now shown. This figure has also now been moved to the Supplementary.

7. Page 15 – 'Participants...' missing with.

This has now been corrected.

8. Page 15 – What does 'young-onset older-onset' mean?

We use the mean of the bimodal distribution (approx. 40 years) shown in Supplementary Figure 1 to classify seminomas as young-onset (occurring in first peak) or older-onset (occurring second peak). We have now made this point clear in the main text and the Methods (see 3.2 Age distribution).

Reviewer #4 (Remarks to the Author): Expert in cancer genomics and evolution, and HLA-LOH

This is an interesting paper with rigorous analysis of WGS data, including mutational signatures, CNAs, predicted mutation time

Validation of previous known features of TGCT. Molecular timing results showing early whole genome duplication but late driver mutation events. I think this paper overall is more valuable for the genomic data it has generated rather than the results, which do not tell as compelling of a story.

We thank the reviewer for their positive comments.

Comments:

- Biggest caveat: missing RNA-seq, so impossible to say what these genomic differences correspond to phenotypically

Could maybe reanalyze the TCGA RNA-seq data in light of their new results?

We thank the Reviewer for these valuable comments. As the Reviewer is aware, our goal was to present the most comprehensive analysis of the genomic landscape of adult testicular cancer to date, by deep whole-genome sequencing. This led to several unexpected results that we believe are both fundamentally important and novel. This work was carried out within the framework of Genomics England and the NHS Genomic Medicine Service. and unfortunately RNA-seq data was not available. Sample collection across the UK and analysis of the whole-genome data alone has taken eight years.

In the future, being able to interrogate these findings further with transcriptome data would be of high value. This is a critical point to emphasise, and we have added a comment in the Discussion on the value of follow-up RNA studies that will facilitate explorations of phenotype. Nevertheless, we hope that the Reviewer will appreciate that our work stands on its own to reveal the broad contours of the TGCT mutational landscape.

- Should have included the eight other samples from the previous largest WGS study of TGCT mentioned in the paper: <https://doi.org/10.1038/s41467-022-31375-4> (both Wedge and Murray are co-authors, and data is on EGA)

We initially attempted to include this data. However, the structure of the data obtained from the Behjati study posed significant challenges. These data encompass a large number of microbiopsies with very low coverage (ranging from 15-48x, whereas the Genomics England samples have an average coverage of 109X) derived from a limited number of individuals. Despite our efforts to apply various analytical approaches, such as the Plackett Luce model for relative ordering of mutations, the statistical inference from this dataset, which includes only 7 malignant TGCTs and 1 GCNIS sample, did not yield statistically significant insights.

- Too small/heterogeneous of a cohort, hard to make any significant comparisons

Testicular germ cell tumours are a relatively rare type of cancer, accounting for just 1% of all cancers that occur in men. So while the cohort is relatively small when compared with recent landscape studies of more common cancer types, this is still the largest whole genome data set published to date. Within this cohort we have managed to capture disease heterogeneity - sampling from primary and metastatic cases, seminomas and non-seminomatous germ cell tumours as well as rarer GCTs such as TGCT with somatic malignancy.

- More in depth analysis of differences according to clinical features (age, treatment response, survival if available?)

Any significant differences have been reported in the main text. As mentioned by the reviewer the structure and size of the cohort (while still the largest cohort of its kind) has made finding significant associations linked with treatment response and survival challenging. However we believe this resource will become a critical springboard for larger cohort follow-ups.

Bimodal distribution of age of onset is interesting; a figure highlighting the differences between two groups would be more compelling

We have now updated the Figure to highlight differences between seminomas and NSGCTs.

- Could perform neoantigen pMHC binding affinity prediction, and compare that to HLA LOH

We thank the Reviewer for this interesting suggestion and have now extended the 'Immune Disruption' section of the paper to include this analysis. Please see the second half of section '*HLA Loss enriched in seminomas*'.

- No p -values in any of the plots, most notably 3b. Also in 3b, the boxplots are covering the data points, order should be switched

We have updated this Figure (now moved to Supplementary) to include p -values. Data points are also now shown.

- Said they extracted 4 tumor samples from the same patient, were those sequenced separately? If so, should look into heterogeneity of the tumor

Yes, these samples were sequenced separately. We have added to the main text of the manuscript to describe these findings (see page 13 from line 380 and Supplementary Figure 20). Broadly our data support limited intra-tumor heterogeneity.

- I'm curious if germline HLA homozygosity is also associated with the risk of these tumors and perhaps the age of diagnosis.

This is a very interesting question. We tested this but found no significant associations between HLA homozygosity (using broad assignments or HLA supertypes) and either the age of diagnosis, clinical stage or pathologic stage.

- It would be helpful for the readers if the authors can provide a summarizing diagram highlighting their main findings for each subtype.

While we have not included a summary figure in the revised manuscript we have undertaken significant revisions, restructuring both the manuscript and accompanying figures, both primary and supplementary materials. These revisions were aimed at highlighting the novelty of our findings and improving the cohesion and relevance across the diverse analyses presented. Our focus was not only on improving the flow but also on establishing stronger connections between findings, making for a more coherent narrative that amplifies the significance and interrelation of our results.

Overall, this paper provides a useful overview of the general whole genome landscape of this understudied cancer, but doesn't go into much depth with regards to the biological/clinical relevance (maybe due to the limitations of the data collected).

REVIEWER COMMENTS

Reviewer #2 (Remarks to the Author):

Overall the authors have addressed my concerns. Claims about negative selection have been softened, which is appropriate in my opinion (while quite cautious and fully acceptable, even the remaining comments in the discussion are perhaps a bit overly optimistic - as alternative explanations to me still seem more likely in the end). Efforts to address the other reviewers' concerns have also led to the presentation being generally improved.

I still find it hard to assess whether the WGD classification, and formula used for this, is reasonable despite being used by PCAWG (it obviously relies on the ploidy estimates being accurate) - but agreement with earlier conclusions is reassuring (Shen et al 2018).

While the novelty may in the end still be somewhat limited, I think this is a useful study worthy of being published.

Reviewer #3 (Remarks to the Author):

This manuscript represents the largest set of whole genome sequencing analysis of testicular germ cell tumor (TGCT), on which the investigators have done extensive analyses. As noted in the initial review, the data provide a very comprehensive picture of TGCT from the DNA perspective and will be useful to investigators. The manuscript provides important insight into the disease through exploration and clarification of known phenomenon (e.g. delineating very early WGS initiation, KRAS amplification and co-occurrence of genetic changes). They also provide a comprehensive evaluation of mutational signatures. The focus in the discussion is on these signatures, but could perhaps be broadened as noted below, to improve the interest. Additionally, as the authors note in the abstract, the manuscript provides an essential and helpful resource, although whether any findings are truly novel is debatable.

For the reviewers, the initial submission was quite challenging with multiple 'errors' in the text and figures, in particular in terms of referring to the correct figures, getting them in order, and lack of clarity in the supplemental figures and tables, in particular. The overall impression was one of a sloppy manuscript, which detracted from a thorough scientific review. In the resubmission, the authors have taken the time in the resubmission to correct many of the errors that were pointed out, and the manuscript is much improved. Additional clarity is still needed in various places, as pointed out below.

1. More detailed clinical information needs to be included for the cases individually, including age of diagnosis, disease stage at diagnosis, treatment (presumably all samples are pre-treatment, but it's not entirely clear). The clinical data are described in the methods, but not described. Important for the signature section that discusses post-chemotherapy samples.

2. The number of cases presented with WGS is not entirely clear. Throughout the manuscript, they discuss 60 cases, but the Method states the following 'The number of individual SNVs and indels called across the cohort reported in the text was based on only 55 of 57 genomes wherein low-purity multi-region samples and samples not generated via PCR-free library preparation were excluded.' Thus, it is hard to know how many samples are included in each analysis – the numbers vary in the supplemental text as well.

3. The authors identify eight genes as being somatically mutated using the IntOGen driver discovery method. Frankly, it's hard to believe that the genes in which two variants (not even clear mutations) are found in the sample set are drivers, and why those genes as opposed to other genes with two variants. The IntOGen algorithm seems too promiscuous, and the evidence to support these variants as drivers is limited. This issue was discussed in the prior review, and the reviewers just refer back to their pipeline – needs to be addressed with more thought – is there pipeline correct?

Particularly problematic are SPEN – two missense variants and ambiguous function, KLF4 – one missense and one stop gained and activating function (so not consistent mechanism and not found in any other sample set of TGCT), and EP300 (two missense variants and LoF mechanism – so certainly stretching it without evidence of functionality). CADD is a poor predictor of functionality and so not a commonly used pathogenicity prediction software, so raises concerns. To note, REVEL is much more commonly used, and felt to be a better pathogenicity predictor.

4. The authors have clarified that Supplemental Figure 4 includes only recurrent focal copy number gains and losses. It would be helpful to see the entire GISTIC plot with all of the copy number alterations. Additionally, genes listed in the text are not all noted on the Supp Figure 4 (presumably former Supp Figure 7), which would be helpful, which was noted in the prior review. DMRT1 is noted in the text later on, but not shown (as are other genes). In the prior review, a comment was made that the authors needed to clarify which interactions were due to physical proximity, which does not appear to have been addressed. Additionally, DMRT1 is not included in the interactions shown in Supplemental Figure 4, presumably because is not a tumor suppressor (but why not?).

5. To note in general, the lack of consideration of germline variation as it related to their findings is a missed opportunity. For example, multiple variants are associated with DMRT1 are associated with susceptibility to TGCT, so that it is interesting that it is significantly lost in their data (not mentioned), and might be of interest to evaluate whether the risk variants are preferentially retained. (Similarly, the amplification of BAK1.) Although this level of analysis may be out of scope, as they have already so much, some comment about their findings in relationship to germline susceptibility would be good to include. To note, it's clearly related, with inherited variants in the chromosomal segregation pathway as demonstrated as being important in TGCT susceptibility and likely enables the early WGD observed in their tumors. (It might be more relevant than pointing out a signature seen once or twice.)

6. In the last review, it was commented upon that the non-coding driver section is not compelling – which it continues not to be.... Could be very short or even cut.

7. Supplementary Figure 7 is difficult to read, and impossible to interpret. Some interpretation of the

Figure should be included in the legend. Supplemental Table 13 needs a key, and to link it up with the text, it would be helpful to either put the location in the text or the genes in the Supplemental Table. This section also does not contribute greatly, and can be cut, especially as the chromothripsis is mentioned elsewhere. (BTW, it is not at all clear how Supplementary Tables 14, 15 support the presence of chromothripsis. What does footprints mean in S15?)

8. The authors bring up the relationship of RefSig R5 to BRCA2 mutations, and note recurrent deletions spanning BRCA2 in TGCT. However, no SBS3 signature is observed and ID6 is very rare, so it is unclear whether BRCA2 is in fact contributing to the signatures overall. It would be helpful to discuss this finding in this context; also noted on page 13.

9. In the response to review, the authors say that they have removed the comment about PTMA. It seems not to be correct, as there is discussion about PTMA on Page 15 line 444 – which needs to be addressed. One could argue that rather than the genes identified on page 15 being rare drivers, they are overcalls by their pipeline.

10. Page 15 – the authors state they found a ‘number of binding site alterations’ – that statement is not supported by their data. (It’s not’s clear why they make this statement, as they acknowledge in their response to reviews and in the results that their associations non-coding variants could be a spurious result.)

11. The discussion focuses almost exclusively on the mutational signatures – and since not much was found there, it’s pretty lackluster. The authors could move the discussion about the early WGD and implications as being early in development (currently in results) to the concluding discussion, and include comment on possible interaction between germline variation and somatic mutations.

Minor Comments:

1. Table S7, S8 – something odd happened in Row 3.

2. As brought up in the prior review, many of the Supplemental Tables have columns with abbreviation that need definition. They have done some work to improve the legend for those columns, but many of the Supplemental Tables still need labeling.

3. Page 9, line 239 – What does ‘complex genes’ mean? Complex in what way?

4. Figure 1 – As RAS mutations were included, would it also make sense to include the KRAS amplifications as part of the information?

5. Page 358 – What are the TGCT-specific drivers that the authors refer to?

Reviewer #4 (Remarks to the Author):

EDITORIAL NOTE: This reviewer only submitted confidential remarks to the Editor. The concerns from this reviewer were addressed.

Response to Reviewers:

We greatly appreciate the comments made by the Reviewers. As before, we have responded to each of these in turn and incorporated as many of the suggested changes as possible. We have again revised the main text and supplementary materials in line with these comments. Each comment is addressed below in red. Changes made within the manuscript have similarly been highlighted in red.

Reviewer #2 (Remarks to the Author):

Overall the authors have addressed my concerns. Claims about negative selection have been softened, which is appropriate in my opinion (while quite cautious and fully acceptable, even the remaining comments in the discussion are perhaps a bit overly optimistic - as alternative explanations to me still seem more likely in the end). Efforts to address the other reviewers' concerns have also led to the presentation being generally improved.

I still find it hard to assess whether the WGD classification, and formula used for this, is reasonable despite being used by PCAWG (it obviously relies on the ploidy estimates being accurate) - but agreement with earlier conclusions is reassuring (Shen et al 2018).

While the novelty may in the end still be somewhat limited, I think this is a useful study worthy of being published.

We sincerely appreciate the Reviewer's positive feedback and are grateful for the opportunity to address their concerns.

In line with comments from Reviewers #2 and #3, we have removed the section on negative selection but hope to develop this finding in future work. We have made a greater effort to emphasize the novelty and potential clinical utility of our study.

Ambiguity in estimating whole genome duplications is a difficult problem in copy number analysis but we are confident that the approach established for annotating WGD status by the PCAWG study is sound. As Reviewer #2 points out, our data agree with Shen et al 2018 and more recently Oliver et al 2022 (PMID:35953478). Other supporting evidence include the observation of mostly synchronous gain patterns, expected in tumours with WGD. Further, we manually reviewed all copy number profiles and used multiple QC metrics (e.g. observation of large (subclonal) homozygous deletions) which would alert us to misannotated samples.

Reviewer #3 (Remarks to the Author):

This manuscript represents the largest set of whole genome sequencing analysis of testicular germ cell tumor (TGCT), on which the investigators have done extensive analyses. As noted in

the initial review, the data provide a very comprehensive picture of TGCT from the DNA perspective and will be useful to investigators. The manuscript provides important insight into the disease through exploration and clarification of known phenomenon (e.g. delineating very early WGS initiation, KRAS amplification and co-occurrence of genetic changes). They also provide a comprehensive evaluation of mutational signatures. The focus in the discussion is on these signatures, but could perhaps be broadened as noted below, to improve the interest. Additionally, as the authors note in the abstract, the manuscript provides an essential and helpful resource, although whether any findings are truly novel is debatable.

We are grateful for the Reviewer's positive feedback and the chance to address their concerns. In response to comments regarding our discussion of mutational signatures, we have revised this section to ensure a more balanced exploration of our findings.

Regarding the novelty of our findings, we appreciate the Reviewer's point. Nonetheless, we are confident that our manuscript offers an essential resource for understanding the TGCT genomic landscape, consolidating known observations and offering important insights into underexplored or previously entirely unexplored aspects of TGCT pathogenesis.

For the reviewers, the initial submission was quite challenging with multiple 'errors' in the text and figures, in particular in terms of referring to the correct figures, getting them in order, and lack of clarity in the supplemental figures and tables, in particular. The overall impression was one of a sloppy manuscript, which detracted from a thorough scientific review. In the resubmission, the authors have taken the time in the resubmission to correct many of the errors that were pointed out, and the manuscript is much improved. Additional clarity is still needed in various places, as pointed out below.

We greatly appreciate the thoroughness of the Reviewer's comments and believe that we have addressed these in this further round of revisions.

1. More detailed clinical information needs to be included for the cases individually, including age of diagnosis, disease stage at diagnosis, treatment (presumably all samples are pre-treatment, but it's not entirely clear). The clinical data are described in the methods, but not described. Important for the signature section that discusses post-chemotherapy samples.

Due to the limited size of the cohort, the rarity of the cancer type, and the fact that clinical information has been made available to us via NHS England, we are limited in terms of the types and the extent of individual-level information that can be disclosed in the paper, which is completely outwith our control/jurisdiction. In particular, we are not permitted to include clinical data pertaining to groups of less than five individuals, under rules associated with the Genomics England Research Environment. However, we have made every effort to work within these confines and to identify significant associations between genomic data and clinical data.

Age of diagnosis is described in Figure 1 and in Supplementary Figure 1. We have now added a panel to Figure 1 to provide detailed information about clinical stage and also amended the text

to indicate the number of samples that were collected pre-treatment (page 5, from line 115). We also discuss post-chemotherapy samples with reference to the signature analysis throughout the text (e.g., page 9, lines 261; page 14, line 409).

2. The number of cases presented with WGS is not entirely clear. Throughout the manuscript, they discuss 60 cases, but the Method states the following ‘The number of individual SNVs and indels called across the cohort reported in the text was based on only 55 of 57 genomes wherein low-purity multi-region samples and samples not generated via PCR-free library preparation were excluded.’ Thus, it is hard to know how many samples are included in each analysis – the numbers vary in the supplemental text as well.

The number of samples used varies according to the strict criteria put in place for each analysis. Mostly these criteria relate to how samples were sequenced (PCR-free library prep or not), and whether multi-samples from a single individual were included. We have now added a clear statement outlining these criteria for each analysis performed in the paper in a dedicated section of the Methods (see Section 5 *Sample Selection*).

3. The authors identify eight genes as being somatically mutated using the IntOGen driver discovery method. Frankly, it’s hard to believe that the genes in which two variants (not even clear mutations) are found in the sample set are drivers, and why those genes as opposed to other genes with two variants. The IntOGen algorithm seems too promiscuous, and the evidence to support these variants as drivers is limited. This issue was discussed in the prior review, and the reviewers just refer back to their pipeline – needs to be addressed with more thought – is their pipeline correct?

The current IntOGen pipeline runs seven state-of-the-art driver discovery methods and combines their results. This approach has been in development for the past 10+ years (Gonzalez-Perez et al. *Nat Commun* 2013 PMID: 24037244) and has been applied robustly to tens of thousands of samples (e.g. Pich et al. *Nat Commun* 2022 PMID: 35871184). The current citation for IntOGen (Martínez-Jiménez, et al. *Nat Rev Cancer* 2020 PMID: 32778778) has been cited almost 500 times. While no bioinformatic approach is perfect or infallible, this method is considered as close to a gold standard approach for driver discovery as is currently available.

Particularly problematic are SPEN – two missense variants and ambiguous function, KLF4 – one missense and one stop gained and activating function (so not consistent mechanism and not found in any other sample set of TGCT), and EP300 (two missense variants and LoF mechanism – so certainly stretching it without evidence of functionality). CADD is a poor predictor of functionality and so not a commonly used pathogenicity prediction software, so raises concerns. To note, REVEL is much more commonly used, and felt to be a better pathogenicity predictor.

While we hope to be able to provide further evidence in support of low frequency drivers in future large cohort studies, we are confident that we have provided sufficient evidence to describe putative drivers in this work.

Many recent cancer genomic studies use CADD scores in the way that we have (e.g. Bailey, et al. *Nat Commun* 2020 PMID: 32958763; Sørensen et al. *eLife* 2023 PMID: 36883553, Shuai et al *Nat Commun* 2020 PMID: 32024818). In the latter reference, the PCAWG Drivers and Functional Interpretation Working Group relied entirely on CADD scores to detect drivers in WES data. It should also be noted also that CADD scores are used by OncodriveFML which is just 1 of the 7 driver discovery programs implemented as part of the IntOGen pipeline. For example, *KLF4* was also identified as a putative driver using dNdScv (PMID:29056346). Reports implicating either nonsense mutations in *EP300* or deep deletions of said gene in testicular cancer, albeit in a small number of cases, further support our findings (Tu et al. *Cancer* 2016 PMID: 27018785; Wyvekens et al. *Modern Pathology* 2022 PMID: 36030288).

Lastly, we have been careful throughout to refer to these as putative and not bona fide drivers and hope that these results will be borne out by larger studies.

4. The authors have clarified that Supplemental Figure 4 includes only recurrent focal copy number gains and losses. It would be helpful to see the entire GISTIC plot with all of the copy number alterations. Additionally, genes listed in the text are not all noted on the Supp Figure 4 (presumably former Supp Figure 7), which would be helpful, which was noted in the prior review. *DMRT1* is noted in the text later on, but not shown (as are other genes). In the prior review, a comment was made that the authors needed to clarify which interactions were due to physical proximity, which does not appear to have been addressed. Additionally, *DMRT1* is not included in the interactions shown in Supplemental Figure 4, presumably because is not a tumor suppressor (but why not?).

We appreciate the Reviewer's comment. As per their suggestion, we have included the entire GISTIC plot or raw copy number plot in the Supplementary materials (see Supplementary Figure 3). Additionally, *DMRT1* is now labelled in Supplementary Figure 3. Initially, the peak spanning *DMRT1* was left unlabeled as our annotation was limited to oncogenes and tumour suppressors only. A complete annotation of GISTIC peaks is provided in Supplementary Table 11 and this is noted in the figure legend. It should be noted that due to space constraints, it is not feasible to include all gene names in the plot without compromising legibility.

DMRT1 is not included in the interactions shown in Supplemental Figure 4 because, as the viewer correctly notes, our focus is exclusively on tumour suppressors and oncogenes, which aligns with the standard approach for this type of interaction analysis. Specifically, each peak is labelled with the name of a biologically relevant tumour suppressor or oncogene.

In response to previous comments from the review we now note in the text that only one statistically significant interaction involves an inter-chromosomal or proximal pair (page 7, line 187-188).

5. To note in general, the lack of consideration of germline variation as it related to their findings is a missed opportunity. For example, multiple variants are associated with *DMRT1* are associated with susceptibility to TGCT, so that it is interesting that it is significantly lost in their data (not mentioned), and might be of interest to evaluate whether the risk variants are

preferentially retained. (Similarly, the amplification of BAK1.) Although this level of analysis may be out of scope, as they have already so much, some comment about their findings in relationship to germline susceptibility would be good to include. To note, it's clearly related, with inherited variants in the chromosomal segregation pathway as demonstrated as being important in TGCT susceptibility and likely enables the early WGD observed in their tumors. (It might be more relevant than pointing out a signature seen once or twice.)

While some of the analyses suggested by the reviewer are outside of the scope of current work, germline variation is indeed something we have carefully considered during preparation of the manuscript. In particular, to search for rarer high to moderate impact germline variation, we applied ALFRED (PMID: 35388192), a statistical method that performs two genome-wide tests of Knudson's two-hit model for systematic discovery of germline cancer predisposition genes. ALFRED test 1 identified 16 putative germline predisposition genes (*HIVEP3*, *PBX3*, *OVCH2*, *HKDC1*, *GPRIN2*, *LIPK*, *AC011825.3*, *RNF125*, *ZGRF1*, *MCC*, *SATB1-AS1*, *DLEU1*, *AL359232.1*, *TLE1*, *DPH1*, *PCSK5*) while ALFRED test 2 detected just one gene ($p=0.00672$), *ATP10B* (ATPase Phospholipid Transporting 10B). We were not satisfied that there was sufficient support for these predisposition genes given no previous support in the literature and that no common gene was identified by tests 1 and 2. However, we intend to develop this further in future work.

BAK1 remains unlabelled as although the peak observed in Figure 4a spans *BAK1*, it was not detected as a significant enriched or recurrent event using either corrected permutation testing (or using GISTIC).

6. In the last review, it was commented upon that the non-coding driver section is not compelling – which it continues not to be.... Could be very short or even cut.

We appreciate that there is consistent feedback from the Reviewers that there is not sufficient evidence to support this section of the paper and have accordingly removed it.

7. Supplementary Figure 7 is difficult to read, and impossible to interpret. Some interpretation of the Figure should be included in the legend. Supplemental Table 13 needs a key, and to link it up with the text, it would be helpful to either put the location in the text or the genes in the Supplemental Table. This section also does not contribute greatly, and can be cut, especially as the chromothripsis is mentioned elsewhere. (BTW, it is not at all clear how Supplementary Tables 14, 15 support the presence of chromothripsis. What does footprints mean in S15?)

We have now removed Supplementary Figure 7. We agree that this figure might be difficult to interpret. Instead the relevant information is now described in the Methods (Section 9.4.3. *Identifying SV hotspots*).

We have updated the key for Supplemental Table 13. We also now include the locations in the text.

Supplementary Tables 14 and 15 report results from the ClusterSV tool. We have now highlighted the relevant field of each table which supports the presence of chromothripsis.

8. The authors bring up the relationship of RefSig R5 to BRCA2 mutations, and note recurrent deletions spanning BRCA2 in TGCT. However, no SBS3 signature is observed and ID6 is very rare, so it is unclear whether BRCA2 is in fact contributing to the signatures overall. It would be helpful to discuss this finding in this context; also noted on page 13.

We agree that it is worth noting this context and have updated the main text to engage with this point (see page 11, from line 307).

9. In the response to review, the authors say that they have removed the comment about PTMA. It seems not to be correct, as there is discussion about PTMA on Page 15 line 444 – which needs to be addressed. One could argue that rather than the genes identified on page 15 being rare drivers, they are overcalls by their pipeline.

As noted in our previous set of responses, we removed the comment concerning *PTMA* and potential therapeutic intervention. However, we still report evidence supporting *PTMA* as a putative driver. As stated above, we believe that the pipeline we are using for driver calling is reliable and that there is significant support for this coming from other studies. We are able to support our *PTMA* finding by linking it with another recent study of germ cell tumours where *PTMS*, a homologue of *PTMA*, is implicated in GCT epigenetic remodelling (Oliver et al. *Nat Commun* 2022 PMID: 35953478).

10. Page 15 – the authors state they found a ‘number of binding site alterations’ – that statement is not supported by their data. (It’s not’s clear why they make this statement, as they acknowledge in their response to reviews and in the results that their associations non-coding variants could be a spurious result.)

We agree with the reviewer and have have now removed this section from the paper.

11. The discussion focuses almost exclusively on the mutational signatures – and since not much was found there, it’s pretty lackluster. The authors could move the discussion about the early WGD and implications as being early in development (currently in results) to the concluding discussion, and include comment on possible interaction between germline variation and somatic mutations.

On re-review, we agree with the Reviewer that there is too great an emphasis on the signature findings, although we believe that many of these observations are novel and will be of interest to other researchers. We have rewritten this section to present a more balanced discussion of our findings.

Minor Comments:

1. Table S7, S8 – something odd happened in Row 3.

We thank the Reviewer for spotting this and have now corrected this typo.

2. As brought up in the prior review, many of the Supplemental Tables have columns with abbreviation that need definition. They have done some work to improve the legend for those columns, but many of the Supplemental Tables still need labeling.

Barring column names that are easily interpretable, we have added to the metadata available with each table (e.g. legends, notes, abbreviations). Given that we have used a wide range of tools to produce the various analyses described in the paper, we cannot exhaustively describe or explain every parameter and output from each tool. However, we now provide links within the supplemental tables directing the reader to a thorough description of the parameters provided by the tool creators in the appropriate release documentations.

3. Page 9, line 239 – What does ‘complex genes’ mean? Complex in what way?

We agree that this is unclear as written. Here we refer to genes spanned by ‘complex’ rearrangements called by Amplicon Architect. We have updated the text to make this point clearer (see page 8, line 235) and refer to the supporting Supplementary Table.

4. Figure 1 – As RAS mutations were included, would it also make sense to include the KRAS amplifications as part of the information?

We show this information in Figure 2, aligning with the section of the paper where we describe analysis of amplifications and deletions in the paper.

5. Page 358 – What are the TGCT-specific drivers that the authors refer to?

We agree that this is unclear as written. Here, we mean drivers previously implicated in other cancers and in TGCT. We have amended the text to say instead ‘*Enriched gains span known cancer and TGCT drivers..*’ (page 12, line 361) which should make this point clearer.

Reviewer #4 (Remarks to the Author):

EDITORIAL NOTE: This reviewer only submitted confidential remarks to the Editor. The concerns from this reviewer were addressed.

REVIEWERS' COMMENTS

Reviewer #3 (Remarks to the Author):

The authors have responded to the reviewers, and the manuscript is much improved in terms of readability. The deep data provided will be of interest to readers, specifically in the TGCT field.

Response to Reviewers

Reviewer #3 (Remarks to the Author):

The authors have responded to the reviewers, and the manuscript is much improved in terms of readability. The deep data provided will be of interest to readers, specifically in the TGCT field.

We sincerely appreciate the Reviewer's positive feedback. We have now included an additional discussion outlining the limitations of our study, as raised by the reviewer. These limitations include the need for further support for putative drivers, the limited study size, and the lack of consideration of germline variation.